# The PYRIN domain-only protein POP2 inhibits inflammasome priming and activation

Rojo A. Ratsimandresy[1], Lan H. Chu[1,2], Sonal Khare[1,†], Lucia de Almeida[1], Anu Gangopadhyay[1,2], Mohanalaxmi Indramohan[1], Alexander V. Misharin[3], David R. Greaves[4], Harris Perlman[1], Andrea Dorfleutner[1] & Christian Stehlik[1,5]

Inflammasomes are protein platforms linking recognition of microbe, pathogen-associated and damage-associated molecular patterns by cytosolic sensory proteins to caspase-1 activation. Caspase-1 promotes pyroptotic cell death and the maturation and secretion of interleukin (IL)-1β and IL-18, which trigger inflammatory responses to clear infections and initiate wound-healing; however, excessive responses cause inflammatory disease. Inflammasome assembly requires the PYRIN domain (PYD)-containing adaptor ASC, and depends on PYD–PYD interactions. Here we show that the PYD-only protein POP2 inhibits inflammasome assembly by binding to ASC and interfering with the recruitment of ASC to upstream sensors, which prevents caspase-1 activation and cytokine release. POP2 also impairs macrophage priming by inhibiting the activation of non-canonical IκB kinase ε and IκBα, and consequently protects from excessive inflammation and acute shock *in vivo*. Our findings advance our understanding of the complex regulatory mechanisms that maintain a balanced inflammatory response and highlight important differences between individual POP members.

[1] Division of Rheumatology, Department of Medicine, Feinberg School of Medicine, Northwestern University, Chicago, Illinois 60611, USA. [2] Driskill Graduate Program in Life Sciences, Feinberg School of Medicine, Northwestern University, Chicago, Illinois 60611, USA. [3] Division of Pulmonary and Critical Care, Department of Medicine, Feinberg School of Medicine, Northwestern University, Chicago, Illinois 60611, USA. [4] Sir William Dunn School of Pathology, University of Oxford, Oxford OX1 3RE, UK. [5] Robert H. Lurie Comprehensive Cancer Center, Interdepartmental Immunobiology Center and Skin Disease Research Center, Feinberg School of Medicine, Northwestern University, Chicago, Illinois 60611, USA. † Present address: Jubilant Biosys Ltd, #96, Industrial Suburb, 2nd Stage, Yeshwanthpur, Bangalore 560022, India. Correspondence and requests for materials should be addressed to A.D. (email: a-dorfleutner@northwestern.edu) or to C.S. (email: c-stehlik@northwestern.edu).

To mount an inflammatory response, which is crucial for host defence and tissue repair, against infections and tissue damage, cytosolic pathogen/microbe-associated molecular patterns (PAMPs/MAMPs) and damage-associated molecular patterns (DAMPs or danger signals) are sensed by pattern recognition receptors (PRRs). PRRs belonging to the AIM2-like receptor and Nod-like receptor (NLR) families trigger the assembly of large protein platforms, called inflammasomes[1,2]. The NLRP3 inflammasome has an important function in response to infectious and sterile activators, and its activation depends on a two-step mechanism, referred to as priming and activation[3,4]. Priming of the NLRP3 inflammasome depends on Toll-like receptor (TLR) signalling that induces NF-κB-dependent expression and post-translational modification of NLRP3 and interleukin (IL)-1β. Then, NLRP3 is activated by a variety of PAMPs/MAMPs and DAMPs, including the gout-causing uric acid crystals, which ultimately trigger $K^+$ efflux as a shared signal[3,5,6]. Recruitment of the adaptor protein ASC to NLRP3 nucleates ASC polymerization and subsequently leads to caspase-1 activation through an induced-proximity mechanism[7]. Active caspase-1 then executes the proteolytic cleavage, maturation and release of the pro-inflammatory cytokines IL-1β and IL-18 and induces pyroptotic cell death[1,8,9]. Although caspase-1 is activated by the canonical inflammasome pathway, caspases-4 (mouse caspase-11) and -5 are activated by the non-canonical inflammasome pathway. In contrast to caspase-1 activation in the canonical inflammasome complex, the non-canonical pathway does not rely on receptor- and ASC-mediated sensing of molecular patterns; instead, non-canonical inflammasome-linked caspases bind directly to lipopolysaccharide (LPS) through their caspase recruitment domain, thus causing their activation[10]. Although caspase-1 induces IL-1β and IL-18 maturation as well as pyroptotic cell death, caspases-4 and -5 only induce pyroptosis, which causes the release of IL-1α and HMGB1 (refs 3,4). Pyroptosis also releases extracellular oligomeric ASC particles, which can be opsonized and phagocytized by neighbouring cells to propagate and perpetuate inflammasome responses[11,12]. Upon blocking the NLRP3 inflammasome with POP1, the ASC particle release is prevented[13]. The mechanism for canonical inflammasome complex assembly depends on specific protein–protein interactions, which are mediated by homotypic PYRIN domain (PYD) interactions between PRRs and ASC, and by homotypic CARD interactions between ASC and pro-caspase-1 (ref. 14). The PRR-initiated nucleation of the prion-like, self-perpetuating ASC polymerization into filaments or spheres is a crucial step[7,11,15–17]. Hence, the PYD is essential for inflammasome assembly and caspase-1 activation, but we are just beginning to understand the molecular processes that regulate inflammasome activation and resolution. Since excessive inflammasome activation causes inflammatory diseases, a balanced inflammasome response is crucial for the host[18,19].

We discovered a family of cellular PYD-only proteins (POPs), including POP1 (also known as PYD containing 1 (PYDC1)), POP2 (also known as PYDC2) and POP3 (refs 20–22). These small proteins are composed of only a PYD, and are encoded in the human genome, but are missing from the mouse genome[21,22]. Interestingly, poxviruses encode viral POPs that function as immune-evasion proteins to dampen host immune responses[23,24]. We previously demonstrated that the monocyte/macrophage-specific expression of POP1 prevents ASC-dependent inflammasome responses by binding to ASC and thereby preventing ASC–PRR interactions, which ultimately ameliorates systemic inflammation and auto-inflammatory disease[13]. POP3, which does not bind to ASC, but specifically interacts with the cytosolic DNA sensors AIM2 and IFI16, also blocks inflammasome

activation and promotes type I interferon production in vivo[20]. The third member, POP2, is unique, as it not only binds to ASC to inhibit inflammasome responses, but also inhibits NF-κB activation in vitro[25,26]. However, its inflammasome and NF-κB-inhibiting roles have not been investigated in detail, and the function of POP2 in vivo is still elusive.

Here we report that human POP2 simultaneously blocks NLRP3 inflammasome priming and activation in a macrophage-specific transgenic mouse model, and thereby ameliorates pathogen- and damage signal-induced systemic inflammation in vivo. Our study provides insights into the complex molecular mechanisms that regulate inflammasome activation in humans to limit systemic inflammatory disease.

## Results

**POP2 protects mice from excessive inflammation.** Reminiscent of POP1 and POP3, POP2 is also expressed in $CD68^+$ macrophages in inflamed human lung tissue, as shown by immunohistochemistry using a custom-raised POP2 antibody, which we validated for specificity among POPs[20] (Fig. 1a). In addition, we detected strong POP2 staining in the synovial lining of human rheumatoid arthritis (RA) patient samples, which is rich in $CD68^+$ macrophages and correlates with disease severity[27,28] (Fig. 1a). Interestingly, similar to POP1 and POP3, POP2 is also lacking in mice, and evolved recently in the mammalian genome of humans and closely related primate species[22,29]. To investigate the physiological function of POP2, we generated transgenic mice expressing POP2 ($POP2^{TG}$) from the human CD68 promoter containing the macrophage-specific IVS-1 enhancer, an approach that we validated earlier for POP1 and POP3 to study macrophage inflammasomes[13,20]. Utilizing the CD68/IVS-1 promoter enables high-level transgene expression in monocytes, tissue-resident macrophages and dendritic cells, but not other cell types[13,20,30]. We observed a comparable restricted expression pattern in our $POP2^{TG}$ mice and detected expression of POP2 in blood $CD11b^+$ cells utilizing POP2-specific Nanoflares. Specifically, POP2 was expressed in $Ly6C^+$ classical monocytes and, to a much lesser extent, in $Ly6C^{med}$ and $Ly6C^-$ monocytes (Fig. 1b and Supplementary Fig. 1). POP2 expression was also confirmed in vitro with macrophage colony-stimulating factor (M-CSF)-differentiated peripheral blood-derived macrophages by immunoblot (Fig. 1c). We showed previously that POP2 was capable of inhibiting NLRP3-mediated inflammasome responses using an in vitro reconstitution system[26]. We therefore investigated the in vivo NLRP3 response to monosodium urate (MSU) crystals, which act as a universal DAMP and are responsible for synovial inflammation in gout patients[6,31]. Injection of MSU crystals into the peritoneal cavity of wild type (WT) mice causes neutrophil infiltration and inflammation, which was determined by in vivo image analysis and subsequent quantification of a luminescent myeloperoxidase (MPO) probe, and this response was markedly attenuated in $POP2^{TG}$ mice (Fig. 1d). MSU crystal-induced neutrophil recruitment depends on the NLRP3–inflammasome-mediated production of IL-1β (ref. 32). Accordingly, IL-1β levels in peritoneal lavage fluids of $POP2^{TG}$ mice were significantly reduced (Fig. 1e). However, unlike in $POP1^{TG}$ mice, the inflammasome-independent release of IL-6 and tumour necrosis factor (TNF) was also reduced (Fig. 1e)[13], which is likely caused by its NF-κB-inhibiting activity that has been described in vitro[25,33]. The lipid mediator leukotriene $B_4$ ($LTB_4$) is produced following MSU crystal-induced tissue damage in response to lysosomal damage and promotes reactive oxygen species production that is required for subsequent NLRP3 activation[34]. In agreement with its function upstream of NLRP3

activation, LTB$_4$ production in response to MSU crystals was not altered in POP2$^{TG}$ mice. Hence, POP2 impairs neutrophil infiltration by a mechanism that involves diminished NLRP3–inflammasome-mediated IL-1β production.

Macrophages release polymerized ASC particles through pyroptosis, an inflammasome-dependent mechanism. ASC particles act as damage signals to further propagate and perpetuate inflammatory responses and are present in the serum

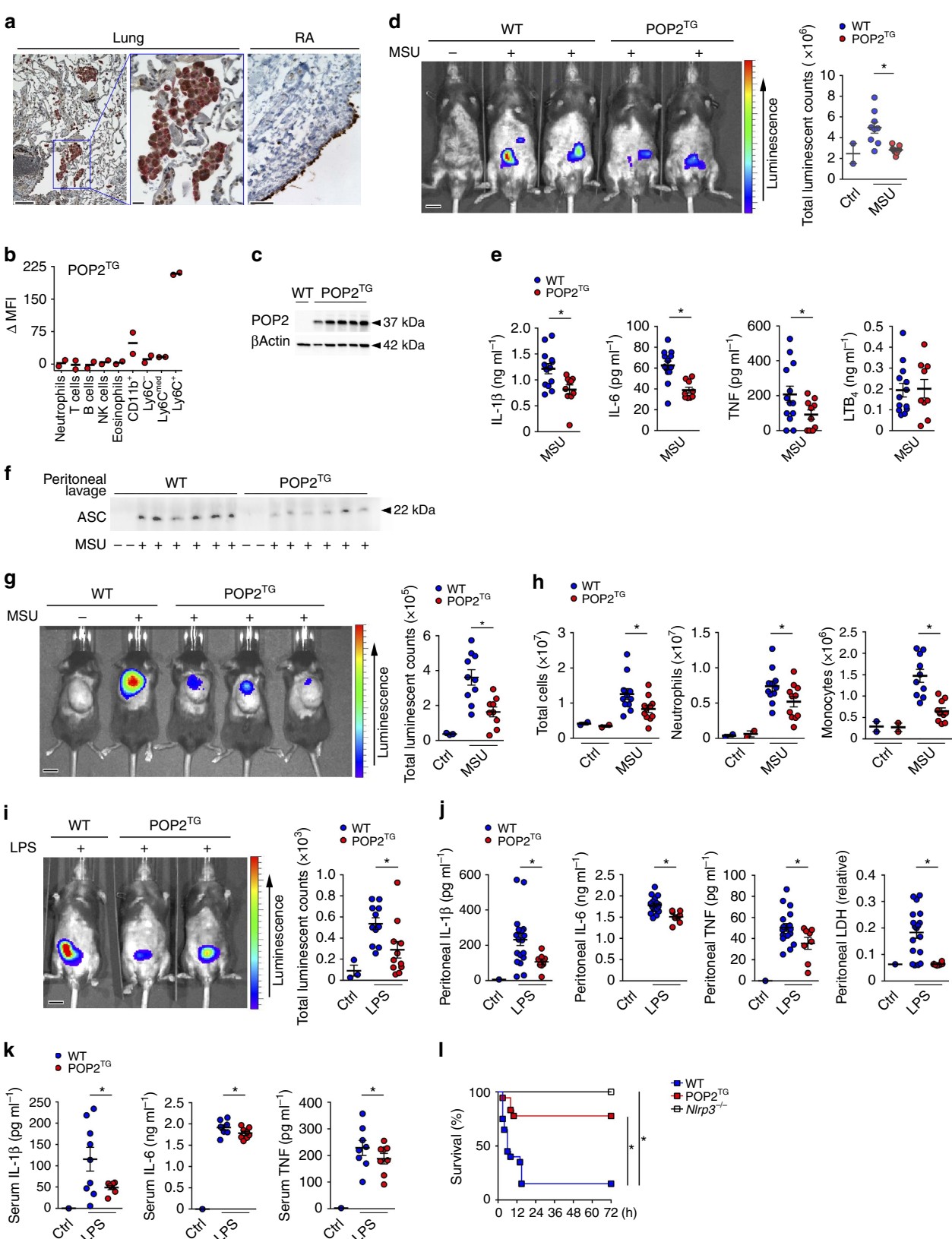

of active Cryopyrin-associated periodic-syndrome patients[11,12]. We recently demonstrated that POP1 expression attenuates the release of ASC particles[13]. Similarly, POP2[TG] mice had reduced extracellular ASC particles compared to WT mice in the cleared peritoneal lavage fluid, as determined by immunoblot analysis (Fig. 1f). In another mouse model that better recapitulates a local inflammatory MSU response in a synovium-like subcutaneous 'airpouch'[35], POP2[TG] mice also showed strongly reduced MPO activity (Fig. 1g). Accordingly, we also observed a reduced number of total infiltrating leukocytes and, in particular, fewer neutrophils and Ly6C[+] inflammatory monocytes (Fig. 1h and Supplementary Fig. 2). To further investigate whether the effect of POP2 was limited to localized MSU crystal responses and DAMPs, we analysed the systemic response to intraperitoneal (i.p.) injection of bacterial LPS, which requires canonical NLRP3 and non-canonical inflammasomes[36,37]. LPS induced a potent neutrophil influx into the peritoneal cavity in WT mice, but this response was strongly reduced in POP2[TG] mice (Fig. 1i). Concomitantly, IL-1β, IL-6 and TNF as well as LPS-induced cell death measured by lactate dehydrogenase (LDH) activity in cleared peritoneal lavage fluids were all reduced in POP2[TG] mice (Fig. 1j). This reduced cytokine response was not limited to the peritoneal cavity, but was also notable on a systemic level, since serum IL-1β, IL-6 and TNF were also reduced in POP2[TG] mice (Fig. 1k). Although the canonical NLRP3 inflammasome contributes to LPS-induced shock[37], it is primarily mediated by the non-canonical inflammasome[36]. To better determine the contribution of POP2 during acute shock, we developed an NLRP3-dependent model. We primed mice with a low-dose LPS before inducing acute shock by i.p. injection of nigericin, which is a microbial toxin derived from *Streptomyces hygroscopicus*. This shock model absolutely depends on NLRP3, since all $Nlrp3^{-/-}$ mice survived, but only 15% of WT mice survived past 15 h. POP2[TG] mice were significantly protected since 78% of mice were still alive and healthy after 72 h, which was the termination point of this study (Fig. 1l). Taken together, our experiments clearly demonstrate that POP2 expression in monocytes and macrophages significantly ameliorates localized and systemic NLRP3 inflammasome responses to DAMPs, MAMPs and toxins *in vivo*. Interestingly, and in contrast to similar experiments performed with POP1 and POP3 transgenic mice, monocyte/macrophage-specific expression of POP2 also reduces local and systemic expression of TNF and IL-6. In addition, POP2 potently prevented death during NLRP3-mediated acute shock.

**Expression of POP2 in mouse macrophages inhibits caspase-1.** POP2 binds to the PYD of ASC in human cells (Fig. 2a)[26], and as expected from the high degree of homology between human and mouse ASC and the apparent phenotype that we observed upon

POP2 expression in mice, human POP2 also co-purified murine ASC in immortalized bone marrow-derived macrophage (iBMDM; Fig. 2a). ASC polymerization is nucleated by active NLRP3 through PYD–PYD-mediated interactions and is a prerequisite for the subsequent proximity-induced caspase-1 activation. Experimentally, the ASC polymerization step can be captured by non-reversible crosslinking of total cell lysate (TCL) and the detection of monomeric and oligomeric ASC by immunoblotting[38]. We utilized the potent and highly specific NLRP3 activator nigericin in LPS-primed BMDM[7,37] to induce ASC polymerization and found that polymerized ASC, but not soluble ASC, was markedly reduced in LPS-primed POP2 BMDM compared to WT BMDM (Fig. 2b). This indicated that POP2 was capable of impairing NLRP3 inflammasome function, which was corroborated by the fact that nigericin-induced caspase-1 activation was significantly blunted in POP2 BMDM compared to WT BMDM, as determined by a caspase-1 fluorochrome-inhibitor of caspases (FLICA)-based flow cytometry assay (Fig. 2c and Supplementary Fig. 3). POP2 expression reduced nigericin-mediated caspase-1 activation from 19.6 to 5.4% of cells, which is only slightly less potent than the 1.4% of cells that we observed in $Asc^{-/-}$ BMDM. We obtained a comparable result for NLRP3 activation by extracellular ATP (Fig. 2d and Supplementary Fig. 3). In addition, the extracellular release of the active p10 subunit of caspase-1 and IL-1β (refs 39,40) in response to LPS priming and MSU stimulation was also reduced in POP2 BMDM compared to WT BMDM (Fig. 2e). Interestingly, under these conditions and at this time point we did not observe any significant reduction of IL-1β expression in TCLs of POP2 BMDM compared to WT BMDM, which indicates that the effect of POP2 on inflammasome-mediated processing of IL-1β is more pronounced than its NF-κB-inhibiting activity at these early time points (Fig. 2e). In addition, POP2 expression ameliorated the IL-1β release from BMDM in response to NLRP3 and AIM2 inflammasome activation with ATP or nigericin and poly(dA:dT), respectively (Fig. 2f,g). $Asc^{-/-}$ BMDM showed no detectable IL-1β release in response to ATP (Fig. 2f). The POP2-mediated inhibition of IL-1β release was not restricted to soluble activators, but was also observed in response to the crystalline NLRP3 activators MSU, calcium pyrophosphate dehydrate (CPPD) and silica (SiO$_2$; Fig. 2h). This effect was observed in two independent POP2[TG] lines underscoring its specificity and excluding its nonspecific effects from transgene integration-derived gene disruption (Fig. 2h). POP2 not only inhibited IL-1β release through the NLRP3 and AIM2 inflammasome, but also other ASC-dependent inflammasomes, including the Pyrin inflammasome activated with the Rho-glucosylating toxin TcdB from *Clostridium difficile*, the NLRC4 inflammasome activated with flagellin, and the NLRP1b inflammasome activated with

**Figure 1 | POP2 expression ameliorates inflammation *in vivo*. (a)** Immunohistochemical staining of CD68 in red and POP2 in brown in inflamed human lung tissue (left and middle) showing the original magnification ×10 (left, scale bar, 50 μm) and ×40 (middle, scale bar, 10 μm) and human RA synovial lining showing the original magnification ×10 (right, scale bar, 50 μm). **(b)** Analysis of POP2 expression by flow cytometry in peripheral blood cell populations from WT and POP2[TG] mice using SmartFlares ($n = 2$). **(c)** POP2 expression in peripheral blood-derived macrophages isolated from WT and POP2[TG] mice by immunoblot. Membranes were stripped and re-probed with a β-Actin antibody as loading control ($n = 5$). **(d)** *In vivo* imaging of MPO activity of infiltrating neutrophils with a luminescent probe in response to i.p. injection with MSU crystals (3 mg; +) or PBS (−) in WT and POP2[TG] mice after 4 h (left, scale bar, 1 cm), and quantification of the luminescent counts (right; $n = 9$). **(e)** Cytokine ELISA and **(f)** immunoblot for extracellular ASC particles in cleared peritoneal exudates of WT and POP2[TG] mice 8 h after MSU injection ($n = 10$–13). **(g)** *In vivo* imaging of MPO activity correlating to MSU-induced neutrophil infiltration into air pouches 7 h after MSU (3 mg per airpouch; +) or PBS (−) injection in WT and POP2[TG] mice (left, scale bar, 1 cm) and signal quantification (right; $n = 9$–10). **(h)** FACS analysis of infiltrating cells in airpouch lavage exudates of WT and POP2[TG] mice 8 h after MSU injection ($n = 11$). **(i)** *In vivo* imaging of MPO activity in LPS-induced peritonitis 4 h after i.p. injection of LPS (2.5 mg kg$^{-1}$) or PBS (left, scale bar, 1 cm) and quantification (right; $n = 11$). **(j,k)** Cytokine ELISA and LDH quantification of **(j)** cleared peritoneal exudates and **(k)** serum of WT and POP2[TG] mice 4 h after LPS injection ($n = 8$–19). **(l)** Survival analysis of the indicated genotypes in response to acute nigericin-induced shock following i.p. injection of nigericin (6 mg kg$^{-1}$) in LPS-primed mice (0.4 mg kg$^{-1}$; $n = 4$–17) and significance was determined by asymmetrical log-rank Mantel–Cox survival test. A standard two-tailed unpaired *t*-test was used for all other calculations. Error bars represent s.e.m., *$P < 0.05$.

lethal toxin from *Bacillus anthracis* (Fig. 2i)[41–45]. Secretion of IL-18 in response to nigericin, poly(dA:dT) and MSU, which is also dependent on caspase-1, was similarly reduced in POP2 BMDM (Fig. 2j). Caspase-1 activation not only promotes cytokine release,

but also induces pyroptotic cell death, which results, among others, in the release of LDH and IL-1α (ref. 46). We observed reduced LDH release in POP2 BMDM and $Asc^{-/-}$ BMDM compared to WT BMDM upon nigericin treatment and DNA

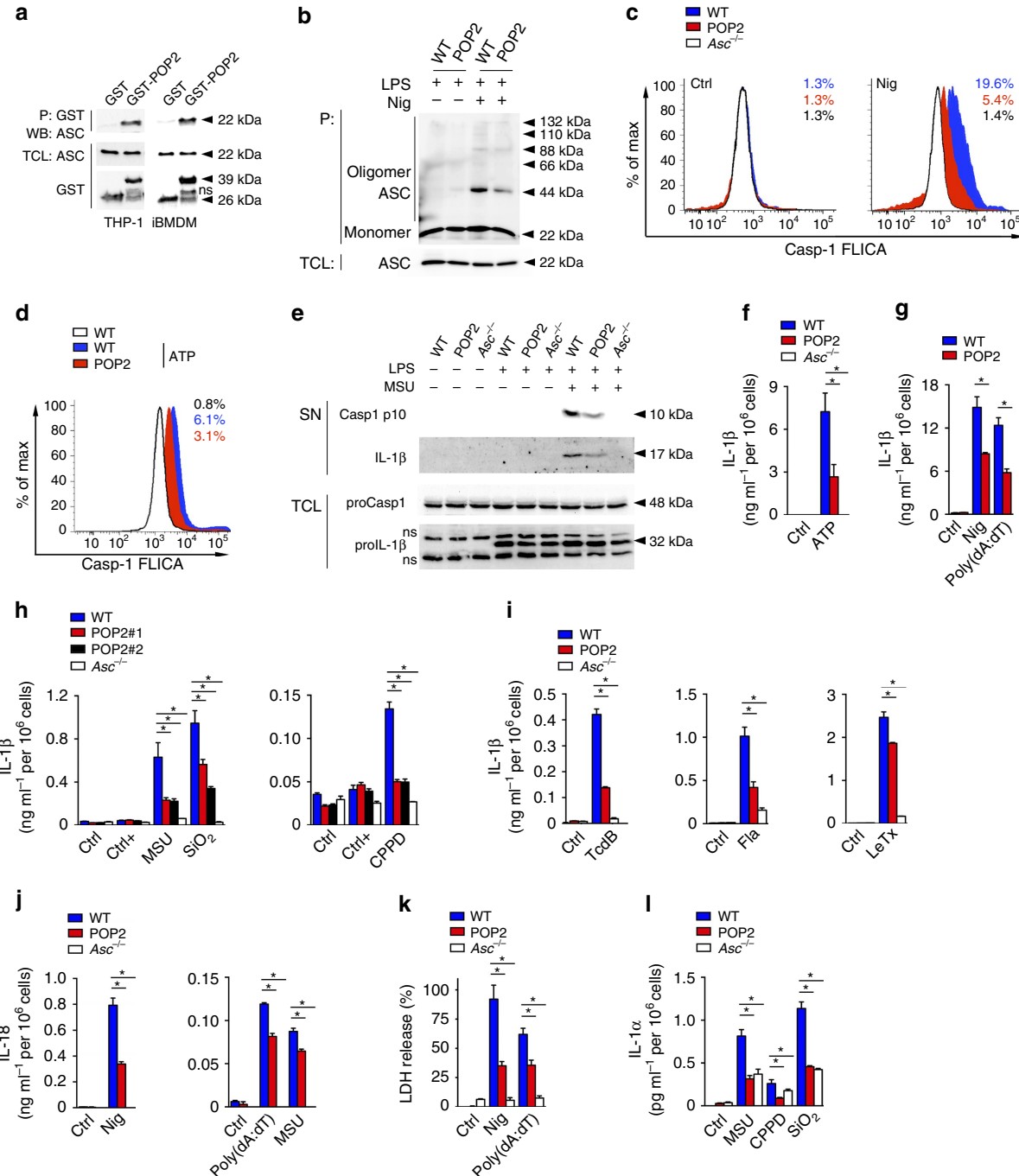

**Figure 2 | POP2 inhibits inflammasome assembly and activation in mouse macrophages.** (**a**) Interaction of GST-POP2 with endogenous ASC from LPS-primed THP-1 cells and iBMDM TCLs using GST as negative control and showing 10% TCL as input. (**b**) Immunoblot analysis of ASC polymerization (oligomer) in untreated or LPS/nigericin (Nig)-treated WT BMDM and POP2 BMDM after non-reversible crosslinking of pellets (P) and in TCL. (**c,d**) Flow cytometric quantification of active caspase-1 in LPS-primed WT, POP2 and $Asc^{-/-}$ BMDM that were treated with control (Ctrl) or (**c**) nigericin or (**d**) ATP, % FLICA$^+$ live, single cells is listed. (**e**) Immunoblot analysis of active caspase-1 p10 and IL-1β release into culture SN of control, LPS-primed and LPS-primed and MSU-treated WT, POP2 and $Asc^{-/-}$ BMDM. Pro-caspase-1 and pro-IL-1β expression in TCL confirms equal loading. ns indicates a cross-reactive nonspecific protein. (**f–i**) Analysis of culture SN for IL-1β release by ELISA in control (Ctrl), LPS-primed (Ctrl+), or LPS-primed and (**f**) ATP-treated, (**g**) nigericin-treated or poly(dA:dT)-transfected and (**h**) MSU, SiO₂ or CPPD or (**i**) TcdB-treated or flagellin (Fla) and *Bacillus anthracis* lethal toxin-transfected WT, POP2 and $Asc^{-/-}$ BMDM. (**h**) POP2#1 and POP2#2 BMDM represent two independent POP2$^{TG}$ lines. (**j–l**) Analysis of culture SN for release of (**j**) IL-18, (**k**) LDH and (**l**) IL-1α in response to the treatment of WT, POP2 and $Asc^{-/-}$ BMDM as above. Between three and five independent repeats were performed for each experiment. Significance was calculated by a standard two-tailed unpaired *t*-test and error bars represent s.e.m., *$P < 0.05$.

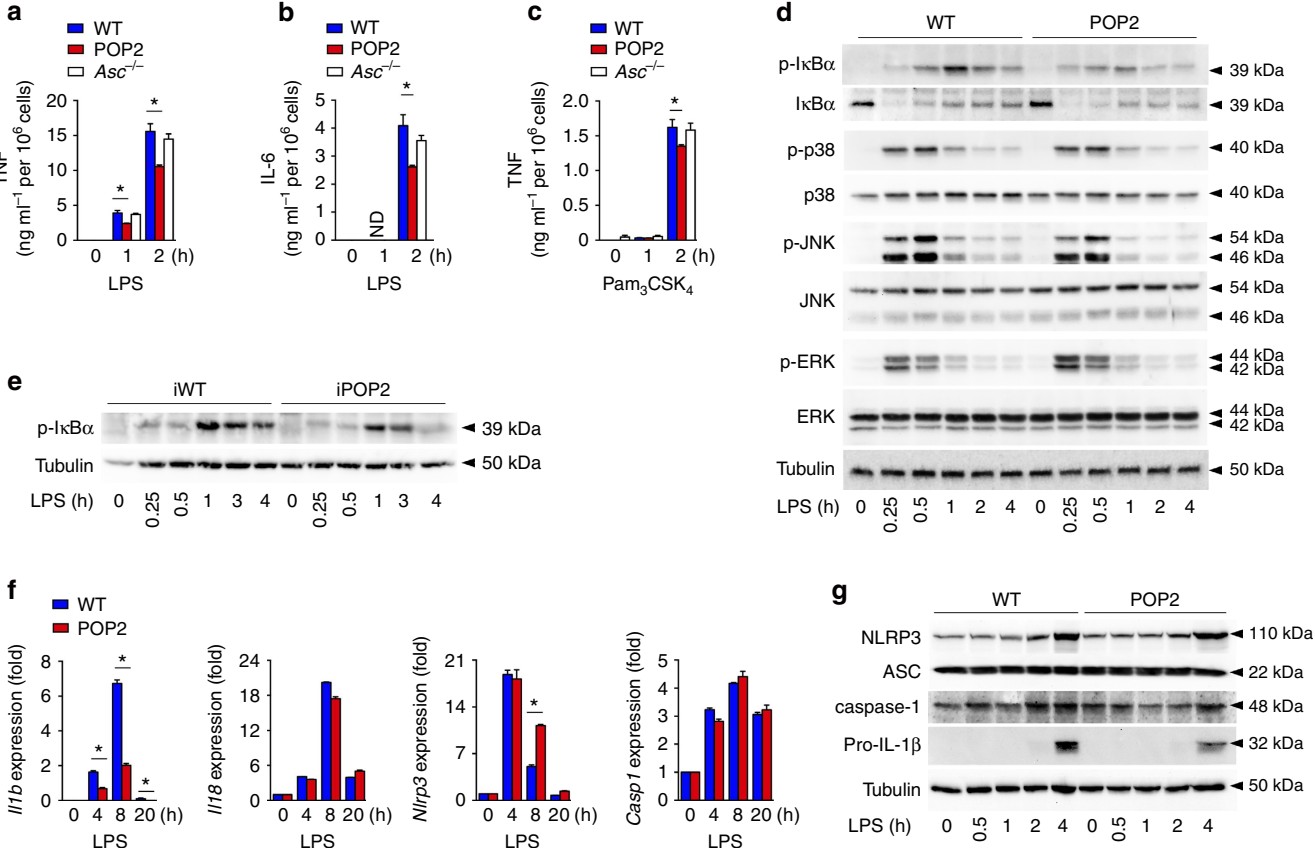

**Figure 3 | POP2 modulates NF-κB-mediated priming of mouse macrophages.** (**a–c**) Analysis of culture SN for (**a,c**) TNF and (**b**) IL-6 release by ELISA in response to (**a,b**) LPS or (**c**) Pam₃CSK₄ treatment of WT, POP2 and $Asc^{-/-}$ BMDM for the indicated times. (**d,e,g**) Immunoblot analysis of phosphorylated (p) proteins and inflammasome proteins in response to LPS treatment of WT and POP2 (**d,g**) BMDM and (**e**) iBMDM for the indicated times. Tubulin protein levels were used as a loading control. ns indicates a cross-reactive nonspecific protein. (**f**) Real-time PCR analysis of *Il1b*, *Il-18*, *Nlrp3* and *Casp1* transcripts in WT and POP2 BMDM in response to LPS treatment for the indicated times presented as fold compared to untreated WT BMDM. Three independent repeats were performed for each experiment. Significance was calculated by a standard two-tailed unpaired *t*-test and error bars represent s.e.m., *$P < 0.05$.

transfection (Fig. 2k) and reduced IL-1α release upon MSU, CPPD and SiO₂ treatments (Fig. 2l). Overall, our results show that POP2 inhibits ASC oligomerization, caspase-1 activation, cytokine release and pyroptosis, which together can explain the ameliorated inflammatory responses observed in POP2^TG mice *in vivo*.

**POP2 inhibits inflammasome priming in mouse macrophages.** NLRP3 activation requires TLR-mediated priming of macrophages, and since POP2 has been implicated in inhibiting NF-κB activation *in vitro*[25], we assessed the contribution of POP2-mediated modulation of macrophage priming on NLRP3 inflammasome activation. LPS priming of BMDM caused inflammasome-independent release of the pro-inflammatory cytokine TNF, and this response was therefore not affected in $Asc^{-/-}$ BMDM. However, POP2 BMDM showed a modest, but reproducible and significant reduction in the release of TNF (Fig. 3a) and IL-6 (Fig. 3b) in response to LPS. This response was not limited to TLR4 activation by LPS, but was also observed in response to the TLR2 agonist Pam₃CSK₄ (Fig. 3c). As TNF and IL-6 production are regulated by NF-κB through the phosphorylation and degradation of IκBα, we investigated LPS-mediated IκBα phosphorylation. Concurrent with the reduced cytokine release and the previously observed reduced p65 nuclear translocation[25], we observed slightly reduced IκBα phosphorylation in POP2 BMDM compared to WT BMDM after

LPS treatment (Fig. 3d). However, other key MAPK-induced signalling pathways, that is, p38, JNK and ERK were not significantly affected (Fig. 3d). Similar to POP2 BMDM, we found reduced IκBα phosphorylation in iBMDM stably expressing POP2 compared to WT iBMDM, establishing that the effect we observed is not due to transgene integration-mediated gene disruption (Fig. 3e). Next, we investigated the consequence of POP2 expression on the transcription of inflammasome components and cytokines by quantitative PCR (qPCR). In agreement with its NF-κB-inhibitory effect, LPS-induced transcription of *Il1b* was reduced in POP2 BMDM compared to WT BMDM, but transcription of *Il-18* and *Casp1* was not affected (Fig. 3f). Although *Nlrp3* expression is known to be upregulated in response to LPS in an NF-κB-dependent manner[47], we did not observe reduced *Nlrp3* expression in POP2 BMDM, nor even slightly enhanced *Nlrp3* mRNA expression levels. However, NLRP3 protein expression was not significantly altered in POP2 BMDM compared to WT BMDM, nor were ASC or caspase-1. In agreement with our mRNA results, pro-IL-1β protein levels were also reduced (Fig. 3g). Overall, our data demonstrate that POP2 inhibits inflammasome priming in addition to inflammasome complex formation, which likely contributes to the inhibition of inflammasome responses *in vivo*. Hence, POP2 has a unique and fundamentally different function in inflammasome regulation compared to POP1 and POP3, which do not affect NF-κB signalling[13,20].

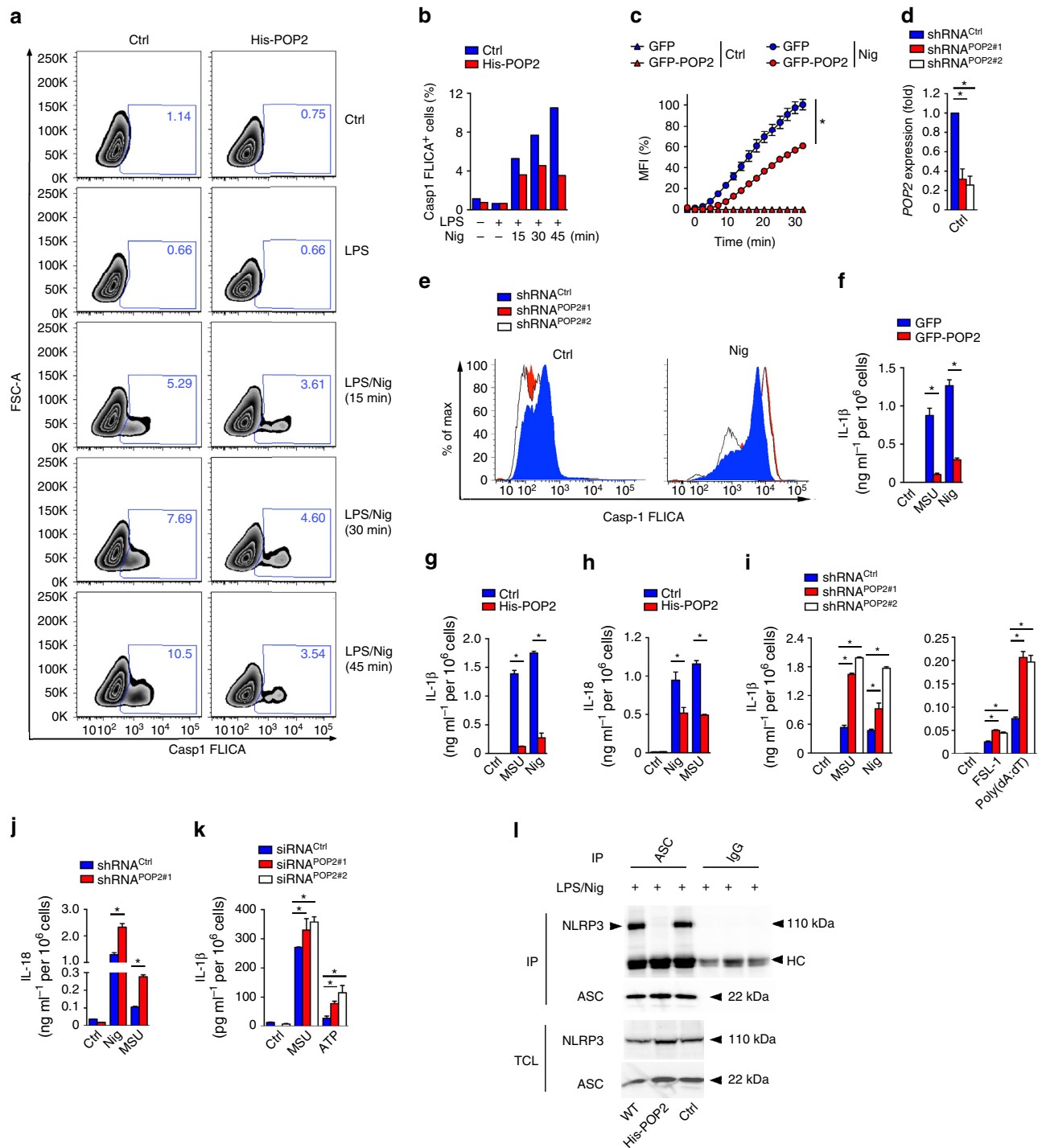

**Figure 4 | POP2 inhibits inflammasome activation in human macrophages.** (**a**,**b**) Flow cytometric quantification of active caspase-1 in control (Ctrl) and His-POP2 stably expressing THP-1 cells in response to nigericin (Nig) treatment for the indicated times in LPS-primed cells, indicating the % FLICA[+] live, single cells showing (**a**) contour plots and (**b**) quantification. (**c**) Kinetic microplate assay of PI uptake by THP-1 cells stably expressing GFP or GFP-POP2 in response to LPS priming followed by nigericin treatment. (**d**) Real-time PCR analysis of *POP2* transcripts in stable control (shRNA[Ctrl]) and two independent POP2 shRNA (shRNA[POP2#1] and shRNA[POP2#2])-expressing THP-1 cells presented as fold compared to control shRNA cells. (**e**) Flow cytometric quantification of active caspase-1 by FLICA assay in LPS-primed POP2-silenced THP-1 cells in response to nigericin (45 min). (**f–j**) Analysis of culture SN for (**f**,**g**,**i**) IL-1β and (**h**,**j**) IL-18 release by ELISA in LPS-primed (**f**) GFP and GFP-POP2-stable THP-1 cells, (**g**,**h**) control and His-POP2-stable cells and (**i**,**j**) control and POP2 shRNA-expressing THP-1 cells in response to the indicated activators. (**k**) Primary human macrophages were transfected with a scrambled control or two POP2-specific siRNAs were primed with LPS, and culture SNs were analysed for secreted IL-1β release in response to MSU crystals and ATP. (**l**) Immunoprecipitation (IP) of ASC or control immunoglobulin G (IgG) from LPS-primed and nigericin-treated WT, control and His-POP2-stable THP-1 cells, followed by immunoblot analysis alongside TCL. HC indicates the antibody heavy chain. Three independent repeats were performed for each experiment. Significance was calculated by a standard two-tailed unpaired *t*-test and error bars represent s.e.m., *P<0.05.

**POP2 has a comparable function in human macrophages**. To ensure that the responses and pathways that we observed in transgenic mice were relevant to human innate immune responses, we investigated inflammasome regulation in human macrophages and the human THP-1 monocytic cell line. We generated cells stably expressing POP2 fused to a 6xHis tag and analysed caspase-1 activation in response to NLRP3 activation with nigericin in LPS-primed cells by FLICA assay. As in BMDM, POP2 THP-1 cells displayed significantly reduced caspase-1 activity compared to control THP-1 cells, which did not increase over time (Fig. 4a,b and Supplementary Fig. 4). Reminiscent of BMDM, stable GFP (green fluorescent protein)–POP2 expression in THP-1 cells also significantly reduced caspase-1-mediated pyroptosis, as determined by kinetic analysis of propidium iodide (PI) uptake by live cells (Fig. 4c). Conversely, stable short hairpin RNA (shRNA)-mediated silencing of POP2 in THP-1 cells using two independent targeting sequences (shPOP2#1 and shPOP2#2), which both reduced POP2 expression by ∼75% (Fig. 4d), increased caspase-1 activation, as determined by FLICA assay (Fig. 4e and Supplementary Fig. 4). We next investigated cytokine release from LPS-primed GFP-POP2-expressing THP-1 cells, which showed markedly reduced IL-1β release in response to MSU crystals and nigericin compared to control cells (Fig. 4f). This response was independent of the fusion tag, as cells stably expressing His-POP2 showed a comparable response (Fig. 4g). In addition to IL-1β, IL-18 secretion was also impaired by POP2 expression (Fig. 4h). Conversely, LPS-primed shPOP2#1 and shPOP2#2 THP-1 cells displayed elevated IL-1β (Fig. 4i) and IL-18 (Fig. 4j) release in response to the NLRP3 agonists MSU crystals and nigericin, the NLRP7 agonist-diacylated lipopeptide FSL-1 and the AIM2 agonists dsDNA analogue poly(dA:dT). To further validate this response, we also silenced POP2 in primary human macrophages by transiently transfecting two independent POP2 short interfering RNAs (siRNAs), which also increased IL-1β release in response to MSU crystals and ATP in LPS-primed cells compared to scrambled control siRNA-transfected cells (Fig. 4k). To further study the mechanism of POP2-mediated NLRP3 inflammasome inhibition, we investigated if POP2 affects the recruitment of ASC to activated NLRP3. Significantly, immunoprecipitation of ASC from LPS-primed and nigericin-treated THP-1 cells co-purified NLRP3 from WT and control THP-1 cells, but not from POP2-expressing cells (Fig. 4l). Control IgG was used as a specificity control and did not purify any NLRP3. Hence, the interaction between POP2 and ASC prevents the recruitment of ASC to activated NLRP3, which ultimately prevents ASC oligomerization, caspase-1 activation and cytokine release in human and mouse macrophages and provides a mechanism for the anti-inflammatory activity of POP2 on ASC-containing inflammasomes.

The NF-κB-inhibitory effect of POP2 observed in BMDM was also recapitulated in human cells since GFP–POP2 THP-1 cells (Fig. 5a) and His-POP2 THP-1 cells (Fig. 5b) secreted less IL-6 in response to LPS, MSU crystals and nigericin than control THP-1 cells, and displayed reduced IκBα phosphorylation in response to LPS (Fig. 5c). Conversely, shPOP2#1 and shPOP2#2 THP-1 cells showed elevated IL-6 levels in response to MSU crystals and nigericin (Fig. 5d) and displayed increased phosphorylation of IκBα in response to LPS (Fig. 5e). This confirms that POP2 functions as an inhibitor of inflammasome priming and activation in mouse and human macrophages. It is unclear how POP2 affects TLR-mediated NFκB activation, but we find that the kinase activity of the upstream canonical IKK complex was not altered in stable POP2-expressing cells after LPS treatment, which was demonstrated by the unchanged or even increased phosphorylation of IKKα/IKKβ and TAK1 (Fig. 5f). However, while LPS-induced phosphorylation of the related non-canonical

IKKε was impaired to a similar level as IκBα, phosphorylation of TBK1, an IKKε-binding kinase, was not altered (Fig. 5f). We observed a similar result in POP2 BMDM compared to WT BMDM (Fig. 5g). Conversely, we observed slightly accelerated and prolonged phosphorylation of IKKε in shPOP2#1 THP-1 cells (Fig. 5h). The role of IKKε in NF-κB signalling is not well established and, while it is sufficient to directly phosphorylate IκBα on S[36] (refs 48–52), it is not necessary for all NF-κB responses, and consequently only a select set of NF-κB-regulated genes is altered in $Ikbke^{-/-}$ mice, which interestingly includes $Il1b$ (ref. 53). The increased NF-κB activity in POP2-silenced THP-1 cells directly translated into increased protein expression of pro-IL-1β, but did not have an impact on caspase-1 and NLRP3 expression (Fig. 5i). However, mRNA levels of $IL1B$ and $TNF$, but not $NLRP3$, were significantly increased (Fig. 5j). Our findings indicate that POP2 interferes with the non-canonical NF-κB activation through IKKε, which contributes to its anti-inflammatory activity.

**POP2 is a late-response regulator of inflammatory responses**. The permanent cellular presence of an inflammasome inhibitor and thereby the constant blocking of the inflammasome would have negative implications on homeostasis[19]. Therefore, we investigated the expression pattern of POP2 in human macrophages and THP-1 cells. POP2 was potently induced in response to LPS in a time-dependent manner, peaking at 20 h in human macrophages (Fig. 6a) and 8 h in THP-1 cells (Fig. 6b), which is reminiscent of POP1 (ref. 13). The relatively late expression of POP2 suggests that it may be involved in the resolution of inflammasome responses. The inducible expression of POP2 was not restricted to LPS-mediated TLR4 activation, but also occurred in response to pro-inflammatory cytokines and interferons (Fig. 6c). Importantly, POP2 also responded to the anti-inflammatory cytokines IL-10/TGFβ and IL-4/IL-13 with elevated expression, which polarize macrophages towards an alternative anti-inflammatory M2c and M2a type, respectively[54], suggesting that POP2 was not only induced by pro-inflammatory stimuli to support resolution of inflammatory responses, but also by anti-inflammatory macrophages to eventually maintain an anti-inflammatory milieu. Although $POP1$ responds very similar to these stimuli, $POP2$ was generally more inducible and was also weakly induced by interferon (IFN)-β, which failed to upregulate $POP1$ (Fig. 6c). In addition, $POP2$ transcripts were increased even further in the presence of an NF-κB inhibitor during TLR2 or TLR4 activation with Pam3CSK4 or LPS in primary macrophages and in THP-1 cells (Fig. 6d,e), while $TNF$ transcripts were reduced as expected (Fig. 6e). Therefore, the ability of POP2 to inhibit NF-κB should elevate its own transcripts, which could ensure that sufficient POP2 levels are maintained during the resolution of inflammatory responses. Furthermore, LPS-inducible $POP2$ expression was also recapitulated in GFP-POP2 THP-1 cells (Fig. 5c) and in BMDM-expressing POP2 from the hCD68/IVS-1 promoter (Fig. 6f), which supported our model system for $in$ $vivo$ studies of POP2. We also found that protein expression levels of POP2 in peripheral blood-derived monocytes (PBMCs) of POP2[TG] mice (Supplementary Fig. 5) inversely correlated to the levels of TNF and IL-1β induced by LPS and MSU, respectively, which substantiated the importance of regulating POP2 expression levels $in$ $vivo$ (Fig. 6g). The correlation between POP2 expression and IL-1β release ($R^2 = 0.4093$) was stronger than between POP2 expression and TNF release ($R^2 = 0.2781$), which might be due to POP2 affecting inflammasome priming and activation for IL-1β release, but only NF-κB-priming activity for TNF release. On the basis of the observed potent inflammasome-

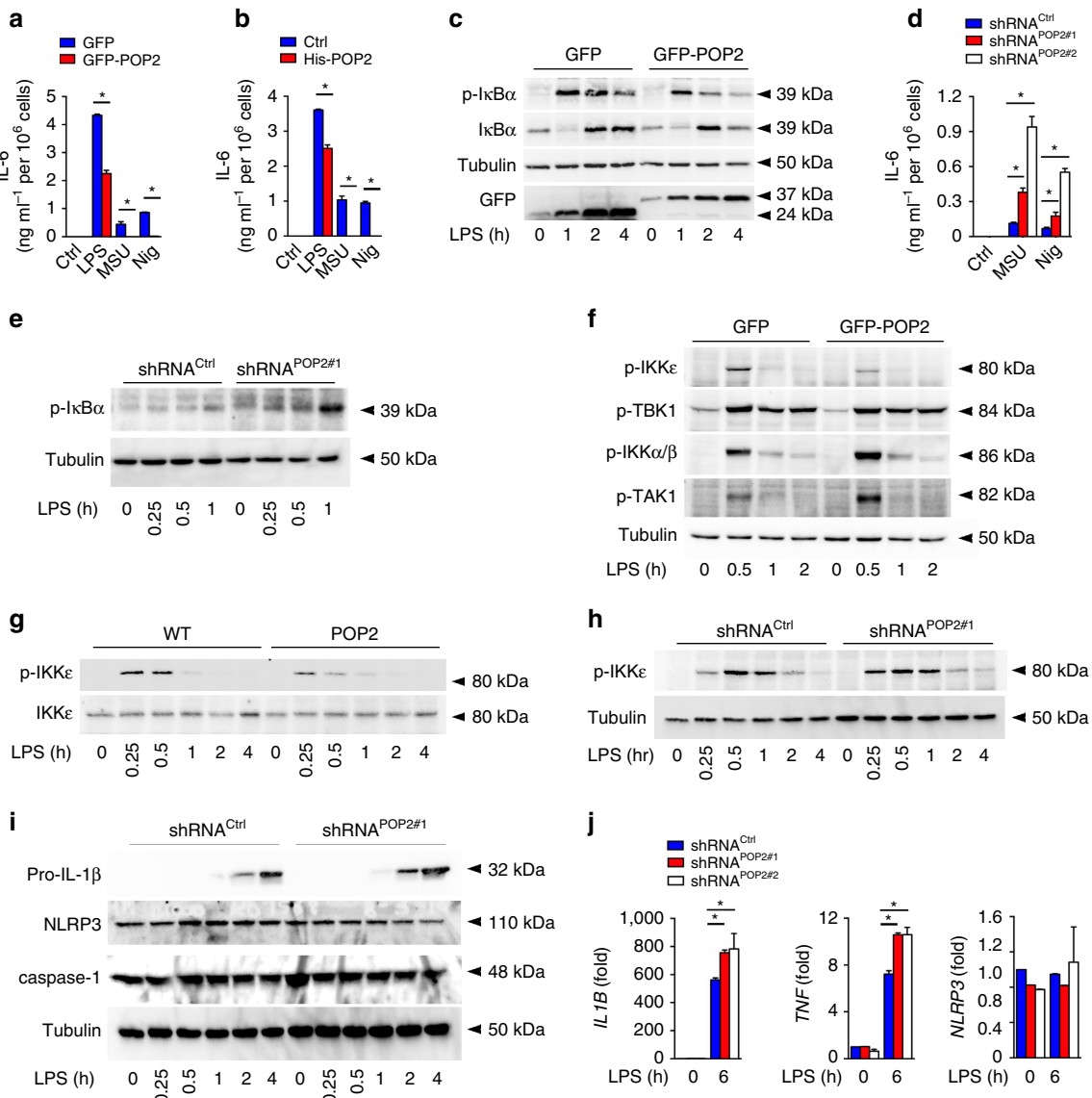

**Figure 5 | POP2 inhibits inflammasome priming in human macrophages.** (**a,b,d**) Analysis of culture SN for IL-6 release by ELISA in response to LPS, MSU crystals and nigericin in (**a**) GFP and GFP-POP2-stable THP-1 cells, (**b**) control and His-POP2-stable THP-1 cells and (**d**) control (shRNA^Ctrl) and POP2 shRNA (shRNA^POP2#1, shRNA^POP2#2) expressing THP-1 cells. (**c,e–i**) Immunoblot analysis of phosphorylated and non-phosphorylated proteins in response to LPS treatment of (**c,f**) GFP and GFP-POP2 (**e,h,i**) control and POP2 shRNA-expressing stable THP-1 cells and (**g**) WT and POP2 BMDM for the indicated times. Tubulin protein levels were used as a loading control and GFP to detect expression of GFP and GFP-POP2. (**j**) Real-time PCR analysis of the indicated transcripts in control shRNA, POP2#1 shRNA and POP2#2 shRNA-expressing THP-1 cells in response to LPS induction presented as fold compared to untreated control shRNA cells. Three independent repeats were performed for each experiment. Significance was calculated by a standard two-tailed unpaired $t$-test and error bars represent s.e.m., *$P < 0.05$.

inhibiting activity of POP2, we tested whether exogenous POP2 could blunt inflammasome responses, and potentially function as a novel class of NLRP3 inflammasome-targeting drug that simultaneously prevents inflammasome priming and activation. Cell-penetrating peptides are frequently used for the intracellular delivery of proteins[55], and we recently demonstrated that TAT-GFP is efficiently taken up by macrophages[13]. We therefore generated recombinant, highly purified cell-permeable POP2 by fusing it with the cell-penetrating HIV TAT peptide (TAT-POP2) and used TAT-GFP as a control. Addition of TAT-GFP to the culture medium did not significantly affect IL-1β release by nigericin in LPS-primed BMDM, while addition of TAT-POP2 significantly reduced IL-1β release (Fig. 6h). The results also showed that our purification approach excluded

endotoxin contamination, as neither TAT-GFP- nor TAT-POP2-primed macrophages caused any IL-1β release, similar to nigericin treatment alone in the absence of priming. Comparable results were obtained from iBMDM (Fig. 6i), demonstrating that intracellularly delivered POP2 can block NLRP3 inflammasome activity.

## Discussion

ASC-containing inflammasomes, and particularly the NLRP3 inflammasome, are crucial for facilitating host defence, wound healing and tissue repair, and for maintaining metabolic health[3,56]. However, deregulated, excessive inflammasome responses also cause acute inflammation and auto-inflammatory

diseases, and therefore a balanced NLRP3 response is essential for maintaining homeostasis[18,19]. Several regulatory factors are known to affect NLRP3 responses. Upstream regulatory events include lysosomal stability, mitochondrial integrity and membrane permeability; direct regulation on NLRP3 include

kinases and deubiquitinases; and downstream signalling events include ASC and caspase-1 modification and activation. However, the key step linking activated NLRP3 to caspase-1 activation is the induced nucleation of ASC polymerization in human and mice, which ultimately drives the induced-proximity activation of

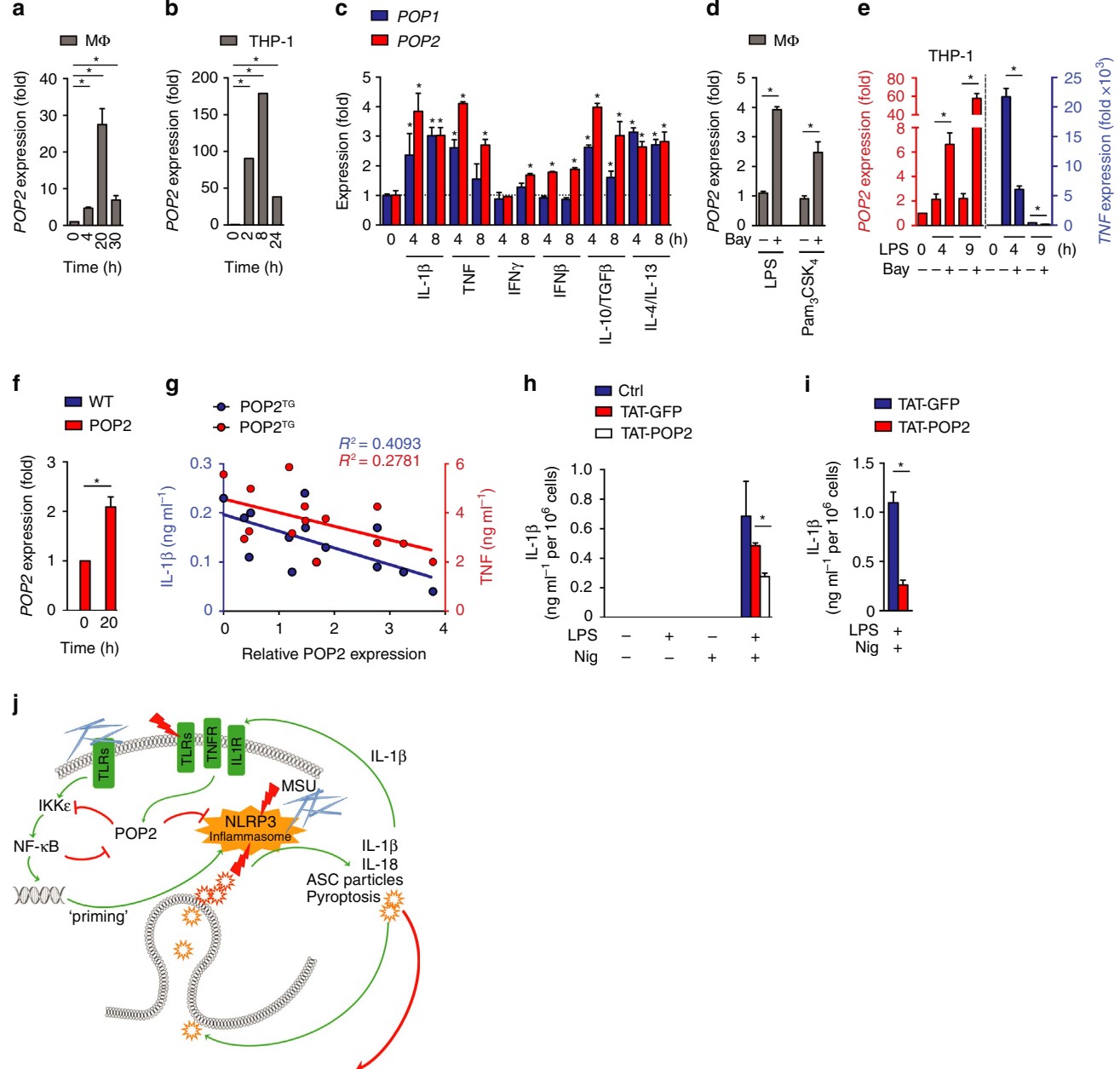

**Figure 6 | POP2 expression is induced in response to pro- and anti-inflammatory cytokines.** (**a–f**) *POP2*, *POP1* and *TNF* transcripts as indicated were measured by real-time PCR in (**a,c,d**) primary human macrophages (MΦ). (**b,e**) THP-1 cells and (**f**) WT and POP2 BMDM treated with various pro- and anti-inflammatory PAMPs and cytokines for the indicated times. (**d,e**) Cells were pretreated with LPS or Pam$_3$CSK$_4$ for 1 h and then further incubated in **d** for a total of 4 h and (**e**) 4 and 9 h in the presence of the NF-κB inhibitor Bay11-7082. (**g**) POP2 expression in BMDM isolated from individual POP2$^{TG}$ mice was determined by immunoblot and densitometric quantification and correlated to the release of IL-1β in response to nigericin or the release of TNF in LPS-primed cells and was analysed by linear regression. (**h,i**) TAT-GFP or TAT-POP2 (0.28 μM) was added to the culture medium of unprimed or LPS-primed (**h**) BMDM and (**i**) iBMDM for 30 min followed by 45 min treatment with nigericin. IL-1β release was quantified by ELISA. (**j**) Proposed function of POP2 as a dual regulator that simultaneously inhibits NF-κB-mediated inflammasome priming and nucleation in macrophages in response to MSU crystals and other activators. POP2 is positioned in a negative feedback loop for inflammasome regulation since it is upregulated by IL-1β, which is blocked upon POP2 expression. For NF-κB signalling POP2 is positioned in a feed-forward regulatory loop since its inhibition of NF-κB leads to POP2 upregulation. Together, POP2 efficiently prevents cytokine release and pyroptosis. Three independent repeats were performed for each experiment in **a–i**. Significance was calculated by a standard two-tailed unpaired $t$-test, and error bars represent s.e.m., *$P < 0.05$.

caspase-1 (refs 7,57). Since this step is mediated by homotypic PYD–PYD interactions, it is not surprising that several proteins regulating specifically this key step belong to the POP family[16,22]. We recently reported that POP1 and POP3 block crucial inflammasome interactions by either binding to ASC or AIM2-like receptors, respectively[13,20]. Here we show that a third POP family member, POP2, also regulates PYD–PYD interactions. While *in vitro* experiments indicated that POP2 regulates PYD–PYD interactions and NF-κB, it had been unknown whether POP2 also regulates these pathways *in vivo*, which is particularly challenging as all POPs are lacking from mice and evolved in higher primates and humans[22,29]. Therefore, we generated a transgenic mouse that expresses POP2 in the monocyte/macrophage lineage, which allowed us to demonstrate the anti-inflammatory function of POP2 *in vivo*, and also its protective function in a newly developed NLRP3-mediated acute shock model using a bacterial toxin. Importantly, we showed that human POP2 functions similarly in primary human macrophages and transgenic mouse macrophages, likely dependent on highly conserved PYDs. In contrast to other POPs, which solely interfere with inflammasome assembly and activation, POP2 also impaired the essential inflammasome-priming step by reducing TLR-induced NF-κB activation. Hence, POP2 has a dual inhibitory activity, which ultimately prevents inflammasome activation, pyroptosis, cytokine and ASC particle release. It is therefore feasible that POP2 has a unique position among NLRP3 regulators, since both activities are combined in a single protein. Future studies will need to address the specific mechanism by which POP2 affects IκBα activation, which is still elusive, as its binding partner ASC does not affect activation of NF-κB in mice[58]. Nevertheless, there is some evidence that suggests that ASC may affect IκBα activation in human cells[59,60]. A POP2 effect on p65 transactivation has been suggested based on expression in HEK293T cells[33], but this would be downstream of IκBα phosphorylation. However, we find that POP2 attenuates IκBα phosphorylation in macrophages. POP2 does not seem to affect the canonical NF-κB pathway, since activation of IKKα/β proceeded normally, but it affects the activation of the non-canonical IKKε. Interestingly, IKKε is sufficient to directly phosphorylate IκBα on S[36] (refs 48–52) and is required for transcription of a select set of NF-κB-regulated genes, including *IL1B*[53]. IKKε is not necessary for all canonical NF-κB responses[53], but its precise role is still incompletely understood.

Although POP2 expression in our transgenic (TG) mice was below the detection level in naive mice, we could detect *POP2* transcripts in monocytes by fluorescence-activated cell sorting (FACS) using Smartflares and POP2 protein in differentiated macrophages by immunoblot. In spite of this low basal expression level, POP2 significantly ameliorated NLRP3-mediated inflammation *in vivo*. We observed a close inverse correlation of *POP2* expression levels and cytokine secretion, which further emphasized the importance of regulating *POP2* expression levels. The inverse correlation between POP2 and IL-1β was stronger than the one between POP2 and TNF, in accordance with the dual inhibitory role of POP2, which blocks *Il1b* transcription by inhibiting NF-κB signalling through reduced IKKε and IκBα phosphorylation as well as the inflammasome-mediated caspase-1 activation responsible for IL-1β maturation and release, while TNF secretion is only blocked on a transcriptional level. *In vivo* we also observed that *POP2* expression mediates a more pronounced reduction in IL-1β than in IL-6 and TNF in response to LPS. However, NF-κB-mediated priming and inflammasome activation are intimately intertwined and difficult to separate. Interestingly, POP2-mediated inhibition of NF-κB signalling occurs late during inflammatory responses but not on the immediate early NF-κB response. In contrast, POP2 affected

caspase-1 activation even after short priming, which suggests a preferential effect on inflammasome activation over transcriptional priming, or that higher POP2 expression levels, after inducible upregulation, are necessary to reduce NF-κB activation. While these POP2-regulated signalling pathways are beneficial for host defence and metabolic and intestinal homeostasis, prolonged activation or defects in timely resolution can result in systemic hyperinflammation and sepsis-like syndromes deleterious for the host. We tested bacterial PAMPs and toxins, and POP2 ameliorated these responses; we, therefore, expect that POP2 would also ameliorate inflammatory responses triggered by bacterial infection. However, a POP2-mediated disruption of early host defence responses could potentially result in an unfavourable elevated bacterial burden. Therefore, the low basal expression of POP2 in conjunction with a late inducible expression of POP2 in response to pro-inflammatory stimuli may reflect a successful mechanism to preferentially dampen the excessive host responses, without impairing early host defence. Significantly, *POP2* was also upregulated by anti-inflammatory cytokines, which polarize towards anti-inflammatory M2 macrophages. Hence, POP2 may not only contribute to the resolution and termination of inflammatory responses, but may also support the maintenance of an anti-inflammatory milieu when expressed in alternatively activated M2 macrophages. Furthermore, we provide evidence that *POP2* transcripts are suppressed by NF-κB, as blocking NF-κB increased *POP2* transcripts. Hence, POP2 is positioned in another regulatory loop, where it inhibits NF-κB, which in turn promotes its own expression. This feed-forward mechanism would ensure that sufficient POP2 levels are maintained during the resolution phase. NF-κB activation usually promotes expression of pro-inflammatory genes and suppresses anti-inflammatory genes. However, a fundamental mechanism of LPS or TLR tolerance during the resolution phase is characterized by impaired NF-κB activation and altered NF-κB dimer composition[61]. Hence, gene expression of the anti-inflammatory *POP2* may be regulated by a comparable mechanism during the resolution phase or during NF-κB p65/p50 and p65/p65 inhibition. Since excessive NLRP3 inflammasome activation is linked to numerous inflammatory diseases, strategies that interfere with this pathway are excellent candidates for ameliorating a wide spectrum of inflammasome-linked disorders. We therefore evaluated a cell-permeable POP2 for its inflammasome-inhibitory activity, which could give rise to novel therapies targeting simultaneously inflammasome priming and activation. Here we substantiate that cell-permeable POP2 mimetics can reduce NLRP3-mediated cellular responses.

In the same issue, Periasamy et al.[62] used a similar approach to determine the *in vivo* role of POP2 by generating POP2[TG] mice under the human POP2 promoter and came to a comparable conclusion that inducible POP2 expression prevented ASC-containing inflammasome-dependent- and inflammasome-independent pro-inflammatory responses. Surprisingly, despite reduced pro-inflammatory cytokines, including IL-18, POP2[TG] mice displayed increased IFN-γ levels, which protected from *Francisella tularensis* and *Streptococcus pneumoniae* infection[62].

It is remarkable that humans/primates have evolved inflammasome-inhibitory proteins, including the POPs[16] that are lacking from mice. It is possible that differentially spliced inflammasome components perform this function in mice, such as ASC splice forms[63], or other, as of yet unidentified POPs are present in the mouse genome that function as homologous or orthologous proteins. Although we have only recently started to appreciate the existence of these proteins in human innate immune cells, we already understand their important role in keeping inflammasome pathways tightly controlled. Our present study demonstrates the unique position of POP2 within the POP

family to restrict inflammasome-mediated inflammatory disease as a dual activity inhibitor *in vivo* (Fig. 6j).

## Methods

**Mice.** pCD68/IVS-1-POP2 was generated by replacing CAT in pCAT-Basic containing the human CD68 promoter and the MΦ-specific IVS-1 enhancer[13,20,30] with GFP-POP2 and flanking the cassette with *Aat*II restriction sites. The *Aat*II fragment was excised, purified and B6.TgN(*CD68-POP2*) TG mice (POP2[TG]) were generated by pronuclear injection into C57BL/6 embryos by the Northwestern University Transgenic and Targeted Mutagenesis Facility. C57BL/6J WT (#000664) and *Casp1/11*[−/−] mice backcrossed for >10 generations onto C57BL/6J (#016621) were obtained from the Jackson Laboratories and C57BL/6J *Nlrp3*[−/−] and *Asc*[−/−] mice from Vishva M. Dixit (Genentech) and described earlier[37,58]. Mice were housed in a specific pathogen-free animal facility, and all experiments were performed on age- and gender-matched, randomly assigned 8–14-week-old mice conducted according to procedures approved by the Northwestern University Committee on Use and Care of Animals. The investigators were not blinded to the genotype of mice.

**Macrophage culture.** Peripheral blood-derived human macrophages were isolated from healthy donor blood (after obtaining informed consent under a protocol approved by the Northwestern University Institutional Review Board) by Ficoll-Hypaque centrifugation and countercurrent centrifugal elutriation and were *in vitro* differentiated[20,64]. Briefly, human PBMCs were isolated by Ficoll-Hypaque centrifugation (Sigma) from healthy donor buffy coats and countercurrent centrifugal elutriation in the presence of 10 μg ml[−1] polymyxin B using a JE-6B rotor (Beckman Coulter). To ensure the purity of PBMCs, cells were washed in Hank's Buffered Salt Solution, resuspended in serum-free RPMI for 1 h, followed by culturing in complete medium supplemented with 20% fetal bovine serum (FBS) for 7 days to differentiate peripheral blood macrophages, which were then cultured in medium supplemented with 10% FBS. Isolated and differentiated peripheral blood macrophages were routinely phenotyped to ensure >85% purity, as determined by flow cytometry for CD45 and CD14. Bone marrow cells were flushed from femurs, differentiated into BMDM with L929 cell-conditioned medium (#CCL-1, American Type Culture Collection (ATCC); 25% *v/v*) or with CMG14-12 cell-conditioned medium (high mM-CSF-producing Ltk[−] cells) obtained from Dr Sunao Takeshita (National Center for Geriatrics and Gerontology Japan)[65] in DMEM medium (#MT-10-013-CV, Fisher Scientific), supplemented with 10% heat-inactivated FBS (Gibco) and 5% horse serum (Gibco) and were analysed after 7 days. PBMCs were differentiated *in vitro* as BMDM, from 150 μl blood for 6 days to determine POP2 expression levels from POP2[TG] mice by immunoblot after normalization with tubulin or βActin expression levels. This method of POP2 quantification was also used for the inverse correlation of POP2 expression levels with inflammatory cytokine production. J2 virus was purified from 72 h-conditioned media of AMJ2-C11 cells (#CRL2456, ATCC) and was used to generate iBMDM[66]. iBMDMs were stably transduced with recombinant lentiviral particles expressing GFP or GFP-POP2 cloned into a modified GFP-expressing pLEX vector. THP-1 cells were obtained from ATCC (#TIB-202) and were routinely tested for mycoplasma contamination using PCR, and all THP-1 cell experiments were performed in the absence of phorbol myristate acetate differentiation. Human macrophages were transfected in 24-well dishes (3.3 × 10[5] cells) with 100 nM siRNA duplexes (F2/virofect; Targeting Systems) and analysed 72 h post transfection (POP2 siRNA#1 sense strand: 5′-CAGAGGUAG-ACAAGGCUAAUU-3′, POP2 siRNA#2 sense strand: 5′-CCCUGGGAAAGGAG-CUACAUU-3′ and Ctrl siRNA F (#sc-44234, Santa Cruz and #SI03650318, Qiagen)[20]. Transfection efficiency was confirmed using qPCR. THP-1 cells were stably transduced with recombinant lentiviral particles using magnetic beads (ExpressMag, #SHM01-1KT, Sigma) and selected with Puromycin (#ant-pr-1, Invivogen). Recombinant lentivirus was produced in Lenti-X-293T cells (#632180, Clontech) by Xfect-based transfection (#631323, Clontech) with pLKO or pLEX and the packaging plasmids pMD.2G and psPAX2 (Addgene plasmids #12259 and #12260), followed by concentration of virus-containing conditioned medium (Lenti-X Concentrator, #631231, Clontech). POP2 shRNA#1: 5′-CCGG-ACAGAGGTAGACAAGGCTAATCTCGAGATTAGCCTTGTCTACCTCTGT-TTTTTG-3′ (TRCN0000253818); POP2 shRNA#2: 5′-CCGGTTCAAGTCTCT-GATCAGAACACTCGAGTGTTCTGATCAGAGACTTGAATTTTTG-3′ (TRCN0000253821) and a non-targeting scrambled control shRNA (Sigma). Human macrophages, THP-1 cells and BMDM were treated for the indicated times with 600 ng ml[−1] *Escherichia coli* LPS (0111:B4, #LPS25, Sigma) or pretreated with 100 ng ml[−1] ultrapure *E. coli* LPS (0111:B4, #LPS-EB Ultrapure, Invivogen), Pam3CSK4 (2 μg ml[−1], #Pam3CSK4, Invivogen), IL-1β (20 ng ml[−1], #IL038), TNF (20 ng ml[−1], #GF023), IL-10 (20 ng ml[−1], #IL010), TGFβ2 (20 ng ml[−1], #GF113), IL-4 (20 ng ml[−1], #IL004), IL-13 (20 ng ml[−1], #IL012), IFNβ (1,500 U ml[−1], #IF014), IFNγ (20 ng ml[−1], #GF305), all from Millipore, CPPD (125 μg ml[−1]; #tlr-cppd, Invivogen), MSU (200 μg ml[−1], see procedure below) or the NF-κB inhibitor Bay11-7082 (20 μM; #196870, Calbiochem/Millipore) for 16 h or as indicated. BMDMs were primed for 4 h with 600 ng ml[−1] *E. coli* 0111:B4 LPS (#LPS25, Sigma-Aldrich), and then transfected with poly(dA:dT) (2 μg ml[−1]; #tlrl-patn, Invivogen), lethal toxin (500 ng ml[−1], #172D and #171D mixed at

a 50/50 ratio, List Labs) or flagellin (100 ng ml[−1]; #tlrl-flic-10, Invivogen) using Lipofectamine 2000 (#11668019, ThermoFisher) as per the manufacturer's instructions, or were stimulated with TcdB (250 ng ml[−1]; #624GT020, Fisher) for 4–16 h. Where indicated, cells were pulsed for 20 min with ATP (5 mM; #A7699, Sigma-Aldrich) or treated for 45 min with nigericin (5 μM; #tlrl-nig, Invivogen).

**MSU crystal preparation.** MSU crystals were prepared by crystallization of a supersaturated uric acid solution as previously described[6,67]. Briefly, uric acid was dissolved in 0.01 M NaOH at 70 °C, adjusted for pH 8 and filtered to 0.22 μM. After 7 days at room temperature, MSU crystals were rinsed twice with ethanol, air-dried and ultraviolet-sterilized.

**Flow cytometry.** FACS was performed as described[13,20]. Briefly, whole blood was collected from anaesthetized animals into EDTA-containing tubes (#41-1504-105, Fisher) and stained with fluorochrome-conjugated antibodies, and erythrocytes were then lysed using BD FACS lysing solution (#349202, BD Biosciences). Cells from the peritoneal cavity were collected after lavage of the peritoneal cavity with 10 ml of ice-cold autoMACS buffer (#130-091-221, Miltenyi Biotec) and were counted using Countess automated cell counter (Invitrogen). Dead cells were discriminated using trypan blue. Cells were stained with live/dead Aqua (#L34957, ThermoFisher) or with the FVD eFluor 506 (#65-0866-14, eBioscience) viability dyes, incubated with FcBlock (#553141, BD Biosciences) and stained with fluorochrome-conjugated antibodies (B220, Horizon PECF594-conjugated, #RA3-6B2, BD Biosciences; CD4, fluorescein isothiocyanate (FITC)-conjugated, #RM4-5, eBioscience; CD8, FITC-conjugated, #53-6.7, eBioscience; CD11b eFluor 450-conjugated, #M1/70, eBioscience; CD11c, PE-Cy7-conjugated, #HL3, BD Biosciences; CD115, phycoerythrin (PE)-conjugated, #AFS98, eBioscience; F4/80, PE-conjugated, #BM8, eBioscience; Ly6G, Alexa Fluor 700-conjugated or PE-Cy7-conjugated, #1A8, BD Biosciences and Biolegend; Ly6C, APC-Cy7-conjugated, #AL-21, BD Biosciences; MHC II (I-A/I-E), PerCPCy5.5-conjugated, #M5/114.15.2, Biolegend; NK1.1, PerCPCy5.5-conjugated, #PK136, BD Biosciences). Data were acquired on a BD LSR II flow cytometer (BD Biosciences). Compensation and analysis of the flow cytometry data were performed using the FlowJo software (TreeStar). 'Fluorescence minus one' controls were used when necessary to set up gates.

**LPS and MSU-induced peritonitis.** Eight- to twelve-week-old male WT and CD68-POP2 mice had their abdomen shaved under anaesthesia, and were randomly selected for i.p. injection with PBS, LPS (2.5 mg kg[−g], *E. coli* 0111:B4, #LPS25, Sigma-Aldrich) or MSU crystals (3 mg). After 4 h, mice were i.p. injected with XenoLight Rediject Inflammation probe (200 mg kg[−1], #760536, PerkinElmer) or luminol (200 mg kg[−1]) from a 50 mg ml[−1] stock solution of Luminol sodium salt (#A4685, Sigma), dissolved in sterile PBS and stored at −20 °C (ref. 68) and *in vivo* bioluminescence was captured by imaging (IVIS Spectrum, PerkinElmer) 10 min post injection with a 5-min exposure on anaesthetized mice[13,20]. Images were quantified with the Living Image software (PerkinElmer). Peritoneal lavages or blood (mandibular bleed) were collected at the indicated time points after PBS, LPS or MSU injection, and cytokine levels were quantified by ELISA.

**MSU airpouch.** Subcutaneous air pouches were generated as previously described[35]. Briefly, anaesthetized 8–14-week-old WT or POP2[TG] mice were injected with 3 and 2 ml of sterile air into the subcutaneous tissue of the back on days 0 and 3. MSU crystals (3 mg) in 1 ml of sterile PBS or 1 ml PBS were injected into the pouch 7 days after the first air injection, and IVIS imaging for MPO activity was performed as described above 7 h later. Pouch lavage was performed 8 h after MSU injection on killed mice by injecting 2.5 ml of PBS containing 5 mM EDTA for analysis of the cellular infiltrate by flow cytometry.

**Nigericin-induced shock.** Mice were primed with LPS (0.4 mg kg[−1] *E. coli* 0111:B4; #LPS25, Sigma-Aldrich), diluted in PBS for 4 h, via i.p. injection, and acute NLRP3-mediated shock was induced by nigericin (6 mg kg[−1], #tlrl-nig, Invivogen) diluted in DMEM via i.p. injection. Survival was followed up to 72 h after nigericin administration.

**Cytokine analysis.** IL-1α, IL-1β (#BMS618/2, IL-6, IL-18, TNF and LTB4 secretion was quantified from clarified culture supernatant (SN) obtained from human macrophages, THP-1 cells, BMDM, mouse serum and peritoneal lavage by ELISA (#559603, #557953, #555240, #555220, #558534, #555212, BD Biosciences, #BMS618/2, #88-5019-86, BMS243/2TENCE, eBiosciences, #DY318-05, R&D Systems, #50-658-126, Enzo). Samples were analysed in triplicates and repeated at least three times, showing a representative result from 10[6] cells.

**Active caspase-1 and pyroptosis assays.** LPS-primed and MSU-treated (200 μg ml[−1]) or nigericin-treated (5 μM for 45 min, unless stated otherwise) cell culture SN were 10% TCA-precipitated and analysed alongside TCL for p45 pro and p10 active caspase-1 by immunoblot analysis. For the active caspase-1 flow

cytometry assay, cells were LPS-primed (1 µg ml$^{-1}$, 30 min) at 37 °C with 5% $CO_2$ and then incubated with biotin-conjugated YVAD-CMK (10 µM, #AS-60841, Anaspec) for 30 min before stimulation with either ATP (5 mM, 20 min) or nigericin (5 µM, 0–45 min). Alexa Fluor 647-conjugated streptavidin (0.25 µg ml$^{-1}$; #S21374, ThermoFisher) was used to quantify active caspase-1 by flow cytometry. LDH release was determined from cleared culture SNs or lavage fluids by colorimetric enzyme activity assay as per the manufacturer's instructions (LDH cytotoxicity detection kit; #MK401, Clontech). Kinetic analysis of PI uptake by live cells following pyroptotic pore formation was carried out as described earlier[69]. Briefly, $10^5$ cells were seeded into black 96-well plates in HEPES-buffered medium (20 mM), supplemented with PI (5 µg ml$^{-1}$; #P4170, Sigma-Aldrich), primed for 1 h with LPS (1 µg ml$^{-1}$) treated with nigericin (5 µM) and PI fluorescence following DNA intercalation was continuously determined every 2 min using a fluorescence plate reader (Synergy HT, Biotek).

**Protein purification and immunoblot assays.** Rabbit polyclonal and mouse monoclonal antibodies to POP2 were custom-raised (keyhole limpet haemocyanin-conjugated EKMNQTHLSGRADE). Rabbit polyclonal antibody to ASC (0.25 µg ml$^{-1}$; #sc-22514-R, Santa Cruz Biotech), mouse monoclonal antibody to ASC (custom), rabbit polyclonal antibody to caspase-1 (0.5 µg ml$^{-1}$; #sc-514, Santa Cruz Biotech), mouse monoclonal antibody to GFP (0.5 µg ml$^{-1}$; #sc-9996, clone B-2, Santa Cruz Biotech), rabbit polyclonal antibody to IL-1β (0.5 µg ml$^{-1}$, #sc-7884, Santa Cruz Biotech), rabbit polyclonal antibodies to IκBα (clone 44D4)/p-IκBα (clone #14D4), JNK (clone #9252)/p-JNK (clone #9251), p38 (clone #9212)/p-p38 (clone #12F8), p42/44 (clone #9102)/p-p42/44 (#9101) and rabbit monoclonal antibodies to p-TBK1 (clone #D52C2), IKKε (clone #2690)/p-IKKε (clone #D1B7), p-TAK1 (clone #90C7) and p-IKKα/β (clone #C84E11; all diluted 1/1,000 and from Cell Signaling Technology) and mouse monoclonal antibody to α-tubulin (0.5 µg ml$^{-1}$, #sc-8035, clone #TU-02, Santa Cruz Biotech), mouse monoclonal antibody to glutathione S-transferase (GST; 0.5 µg ml$^{-1}$, #sc-138, clone #B-14, Santa Cruz Biotech), mouse monoclonal antibody to β-actin (diluted 1/5,000; #A5316, clone #AC-74, Sigma-Aldrich) and mouse monoclonal antibody to NLRP3 (1 µg ml$^{-1}$; AG-20B-0014-C100, clone #Cryo-2, Adipogen) were used for immunoblot analysis.

For phospho-protein analysis, cells were starved for 1 h in a serum-free medium before lysis in 95 °C-preheated Laemmli buffer at different time points, with or without LPS treatment, and proteins were separated by SDS–PAGE, transferred to polyvinylidene difluoride membranes and analysed by immunoblot analysis with the appropriate primary antibodies and horseradish peroxidase-conjugated secondary antibodies (whole-donkey antibody to rabbit IgG (#NA934V, GE Healthcare) and whole-sheep antibody to mouse IgG (#NXA931, GE Healthcare), SuperSignal West Femto Maximum Sensitivity Substrate (#34095, ThermoFisher) and image acquisition (Ultralum).

For co-immunoprecipitation, endogenous NLRP3 inflammasome complexes were purified from ultrapure LPS (100 ng ml$^{-1}$)-primed cells (16 h) after treatment with nigericin (5 µM) for 45 min. Cells were lysed under hypotonic conditions (20 mM HEPES pH 7.4, 10 mM KCl, 1 mM EDTA, supplemented with protease inhibitors) using a 23½ G syringe needle, cleared and adjusted to 50 mM HEPES pH 7.4, 150 mM NaCl, 10% Glycerol, 2 mM EDTA, 0.5% Triton X-100, supplemented with protease inhibitors and subjected to IP by incubating with sepharose-immobilized antibodies as indicated for 16 h at 4 °C, followed by extensive washing with lysis buffer. Bound proteins and TCLs (5%) were analysed by immunoblot.

For GST pull down, POP2 was expressed from pGEX-4T1 and affinity-purified as a GST fusion protein from E. coli BL21 (ref. 26). Protein lysates were prepared from LPS-treated (4 h) iBMDM or THP-1 cells by lysis (50 mM HEPES pH 7.4, 120 mM NaCl, 10% glycerol, 2 mM EDTA, 0.5% Triton X-100, supplemented with protease inhibitors), and cleared lysates were incubated with immobilized GST-POP2 or GST control for 16 h at 4 °C, followed by extensive washing with lysis buffer and analysis by immunoblot.

**ASC crosslinking.** In all, $4 \times 10^6$ BMDM were seeded in 60 mm plates and subjected to crosslinking as described[13,20]. Briefly, cells were either left untreated or were treated with ultrapure LPS (100 ng ml$^{-1}$) for 4 h and with Nigericin for 45 min, culture SNs were removed, cells rinsed with ice-cold PBS and lysed (20 mM HEPES pH 7.4, 100 mM NaCl, 1% NP-40, 1 mM sodium orthovanadate, supplemented with protease inhibitors) and further lysed by shearing. Cleared lysates were stored for immunoblot analysis, and the insoluble pellets were resuspended in 500 µl PBS, supplemented with 2 mM disuccinimidyl suberate (#21655, Pierce) and incubated with rotation at room temperature for 30 min. Samples were centrifuged at 2,300g for 10 min at 4 °C, and the crosslinked pellets were resuspended in 50 µl Laemmli sample buffer and analysed by immunoblot analysis.

**Immunohistochemistry.** De-identified human lung and RA tissue was fixed in 10% formalin, embedded in paraffin, cut into 3 µm sections, mounted, deparaffinized and immunostained with a rabbit polyclonal POP2 (custom-raised) antibody and, where indicated, also with a mouse monoclonal CD68 (#M0876, clone PG-M1, Dako) and peroxidase (horseradish peroxidase)/DAB$^+$ and alkaline

phosphatase/Fast Red enzyme/chromogen combinations (#K5005, Dako) and specific isotype-matched control mouse IgG1 (#X0931, Dako) and rabbit IgG (#X0903, Dako) and haematoxylin counterstaining of nuclei at the Northwestern University Pathology Facility.

**Quantitiative real-time PCR.** Total RNA was isolated from cells using Trizol reagent (#15596018, Invitrogen), treated with DNase I and reverse-transcribed with the Verso cDNA Synthesis Kit (#AB1453A, ThermoFisher). Real-time gene expression analysis was performed on an ABI 7300 Real-Time PCR Machine (Applied Biosystems) and displayed as relative expression compared to β-actin, using pre-designed FAM-labelled mouse primers (Il1b: Mm00434228_m1, Il-18: Mm00434226_m1, NLRP3: Mm00840904_m1, Casp1: Mm00438023_m1) and human primers (IL1B: Hs01555410_m1, TNF: Hs00174128_m1, NLRP3: Hs00918082_m1, POP2: Hs03037798_sH) in combination with VIC-labelled primers for ACTB (Hs99999903_m1 and Mm00607939_s1 for human and mouse samples, respectively) for endogenous controls (all from Invitrogen).

**Cell-penetrating proteins.** 6xHIS-POP2 and 6xHIS-GFP cDNAs were fused with the HIV TAT sequence and purified from E. coli BL21 by sequential NI-NTA affinity (#R901-15, ThermoFisher) and cation exchange chromatography on a High S Support column (#1560030, Bio-Rad) and tested for cellular uptake, as previously described[13]. Cells were pre-incubated with TAT-GFP or TAT-POP2 (0.28 µM) for 16 h, alongside with ultrapure LPS priming (100 ng ml$^{-1}$) or without priming (controls) before activation with nigericin (5 µM, 45 min) and analysis of IL-1β release by ELISA.

**Statistical analysis.** All experiments have been repeated at least three times and in vivo experiments include the indicated number of mice, and graphs represent the mean ± s.e.m. A standard two-tailed unpaired t-test was used for statistical analysis of two groups, with all data points showing a normal distribution with a similar variance between compared groups. Values of $P < 0.05$ were considered significant and survival was calculated by asymmetrical log-rank Mantel–Cox survival test (Prism 6, GraphPad). Sample sizes were selected based on preliminary results to ensure a power of 80 with 95% confidence between populations.

**Data availability.** All data that support the findings of this study are available from the corresponding authors upon request.

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

## Acknowledgements

This work was supported by the National Institutes of Health (AI099009, AR064349 and AI120618 to C.S., AI120625 to C.S. and A.D. and AR066739 to A.D.), a Cancer Center Support Grant (CA060553), the Skin Disease Research Center (AR057216) to C.S. and The British Heart Foundation (RG/15/10/31485) to D.R.G. L.H.C. was supported by the Vietnam Education Foundation Fellowship and the American Heart Association (AHA, 15PRE25700116), S.K. was an Arthritis Foundation fellow (AF161715), L.d.A. was supported by the AHA (11POST585000) and the NIH (T32AR007611 and AR064349S1) and M.I. was supported by the AHA (15POST25690052). We thank all members of the Stehlik lab for helpful discussions. Plasmids pMD2.G and psPAX2 were kindly provided by Didier Trono (École Polytechnique Fédérale de Lausanne, Switzerland), CMG cells by Sunao Takeshita (National Center for Geriatrics and Gerontology Japan) and *Asc*$^{-/-}$ and *Nlrp3*$^{-/-}$ mice by Vishva M. Dixit (Genentech, USA). This work was supported by the Northwestern University Transgenic and Targeted Mutagenesis Laboratory, Pathology core, Center for Advanced Microscopy and Flow Cytometry core facility.

## Author contributions

R.A.R. performed most of the *in vitro* and *in vivo* experiments; L.H.C. generated the stable POP2-expressing and knockdown cells and contributed to their analysis; S.K. performed the NLRP3 and ASC complex analysis and the ASC crosslinking studies; L.d.A. performed the initial characterization of POP2[TG] mice and contributed to the POP2 expression analysis; A.G. supported some of the *in vivo* MSU experiments; M.I. contributed to the *in vivo* experiments and analysis of POP2 transcriptional responses; A.V.M. and H.P. provided support for the design of the flow cytometry analysis; D.R.G. provided the key reagents and advice; R.A.R., A.D. and C.S. wrote the manuscript; R.A.R., A.D. and C.S. conceived of the study and designed the experiments, and A.D. and C.S. provided overall direction.

## Additional information

**Competing interests:** The authors declare no competing financial interests.

