## [Peer Review File · Nature Communications]

Reviewers' comments:

Reviewer #1 (Remarks to the Author):

In this manuscript, Ratsimandresy and colleagues report the generation of a strain of macrophage-specific transgenic mice that expresses human POP2. Studies with this mouse model and human macrophages revealed that POP2 negatively regulates the priming and activation of NLRP3 and AIM2 inflammasomes. As such, POP2 is a novel class of inflammasome inhibitors that may be employed to target pathophysiological conditions involving excessive inflammation.

Previously, the authors employed a CD68 promoter containing macrophage specific IVS-1 enhancer to express POP1 and POP3 in mice. Here the authors used the same strategy to generate a transgenic mouse strain that expresses POP2 only in monocytes, macrophages and DCs. Similar to their in vitro study published previously, the authors showed that the IL-1 secretion downstream of NLRP3 was reduced using monosodium urate crystals as stimulants in the POP2 transgenic mice, concomitant with reduced neutrophil infiltration. Mechanistically, the inhibition of the NLRP3 inflammasome and AIM2 inflammasome was mediated through the direct binding of POP2 with ASC and the reduction of ASC polymerization and subsequent caspase-1 activation.

Surprisingly, the TNF and IL-6 levels were also reduced upon stimulation with either MSU or LPS plus nigericin, suggesting reduction of NF- κ B activation and inflammasome priming by POP2, which is distinct from POP1 and POP3 that do not target NF- κ B. The reduced I κ B and IKK activation appears to correlate with the suppressive effects of POP2 on NF- κ B activation. The above observations in the transgenic mice were confirmed in human THP-1 cells stably expressing POP2.

Because the expression of POP2 was induced by LPS and anti-inflammatory cytokines in a time-dependent manner, which may permit initial protective immune response while facilitating the resolution of inflammation at a later stage, the authors rationalize that POP2 is an important immunoregulatory molecule to ensure balanced inflammatory responses.

Overall this manuscript elucidates the in vivo function of an important regulator that may be relevant to the control of inflammation during infections and a number of autoinflammatory disorders. The important findings should appeal to a broad readership.

One issue that warrants discussion is the effects of POP2 on other members of the inflammasome family. Because POP2 targets ASC, almost all of the inflammasomes may be negatively regulated, including inflammasomes primarily involved in maintaining homeostasis such as the NLRP6 and NLRP12 inflammasomes. How might POP2 impact intestinal homeostasis involving NLRP6 and NLRP12?

Reviewer #2 (Remarks to the Author):

The present study focuses on the unique role of PYRIN domain-only protein POP2 in inhibiting the inflammasome priming and activation in a POP2-transgenic (Tg) mouse model. POP2 ameliorates the inflammatory response in vitro (BMDM) and in vivo (POP2 Tg mice) upon treatment with LPS or monosodium-urate crystals, for PAMP and DAMP signals, respectively. Interestingly, this protection against inflammation also occurs systemically by reducing circulatory levels of IL-1 β and IL-6 and TNF α . THP-1 monocytes and primary human macrophages also show the significant reduction of inflammatory activation in the presence of POP2. ShRNA against POP2 reverses this process and results in the

increased levels of IL-1 β and TNF α . POP2 acts via inhibiting NF- κ B activation and also inhibiting ASC-1 polymerization, thereby preventing Caspase-1 activation and inflammatory cytokines response. This paper represents experimental strategy/tools to elucidate the mechanistic role of POP2 in both in vitro and in vivo conditions. Authors have created a model of POP2 transgenic mouse using a CD68 promoter sequence to study the macrophage inflammasomes.

Major comments:

1. Figure 1I, panel shows MPO activity measured in luminescence units for WT and POP2 Tg mice after LPS treatment. The right panel for POP2 mouse does not show dramatic changes in MPO activity. Did authors check if this particular mouse had similar levels of POP2 expression like other POP2 mice showing lower MPO activity?

2. In Figure 1K, serum IL-6 level does not seem convincing, shown as significantly higher in WT compared to POP-2. It is possible that POP2 can inhibit only secreted serum IL-1 β and TNF α ?

3. Figure 2e, expression of Caspase-1 p10 seems similar between WT and POP2. Figure 2 h and i represent the levels for IL-1 β and IL-18 in response to various other DAMP and PAMP activator molecules. Those differences under CPPD, Poly dA:dT and MSU, do not appear significantly different in WT compared to POP2 probably due to minimal detection. Authors should show these results in a different scale bar graph.

4. In Figure 3, TNF α and IL-6 increase in WT and ASC-/- mice compared to POP2 upon LPS treatment. Why was a different time scale has been chosen for LPS (0,2, and 16h) and Pam3CSK4 (0,1 and 2h) treatment experiments? Also, levels of these cytokines in panel a (2h), b (16h) and c (2h, Pam3CSK4 treatment), seem to be significantly higher for ASC-/- compared to POP2 mice. Have authors looked into this discrepancy?

5. Why are expression levels of p-I κ B α different in POP2 in panel e; (iBMDM) and panel g; BMDM at 1h and 2h time points. In fact, it seems that there is an increased expression of p-I κ B α in POP2 iBMDM after one h of LPS induction, and likewise, an increase in expression of p-I κ B α appears at 2h in POP2 BMDM.

6. Results in Figure 5e contradict the statement by authors, "Conversely, THP-1shPOP2#1 and THP-1shPOP2#2 cells showed elevated IL-6 levels in response to MSU crystals and nigericin (Fig. 5d) and displayed increased phosphorylation of I κ B α in response to LPS (Fig. 5e)". In fact, the p-I κ B α expression is same in shRNA ctrl and shRNAPOP2#1 post LPS treatment ruling out the effect of shRNAPOP2 on the phosphorylation of I κ B α .

7. Authors have shown the regulatory role of POP2 in modulating inflammation in response to PAMPs activators. However, authors should discuss if macrophages expressing POP2 will be efficient in other infection models.

Minor comments:

1. In Materials and Methods, P24, it is stated that peripheral blood-derived monocytes were differentiated in vitro as with BMDM, from 150 μ l blood for six days. Is this correct? It is not clear how many cells are used in this experiment?

Reviewer #3 (Remarks to the Author):

The authors used a POP2 knock-in mouse model to elucidate the *in vivo* protective role of POP2 in damping MAMP and DAMP-induced excessive inflammation. These data were further supported with study results using POP2 stably expressing THP-1 human monocytes. The authors convincingly demonstrated that the protective function of POP2 is mediated by inhibiting inflammasome priming and activation. The study is well designed with proper controls, and the manuscript is well written. However, interpretation of some of the POP2 expression studies (Fig. 5c, Fig 6 f-g) require clarification.

In Fig. 5c, the transcription of GFP and GFP-POP2 mRNA is under the control of pCMV promoter. How does LPS induce both protein expressions? Are these proteins regulated in the post-translational stage by LPS-induced events?

Similarly, in Fig. 6f and g, the promoter of the POP2Tg is CD68/IVS-1 not the endogenous POP2 promoter. Does the POP2 gene expression in the LPS and nigericin treated BMDMPOP2 really reflect endogenous POP2 expression in primates under the treatment condition, or it is likely skewed by the differential activation of CD68/IVS-1 promoter? In addition, if POP2 proteins are also regulated in translational and post-translational stages, the observed correlation between cytokine(IL-1beta, INF-alpha) production and POP2 gene expression may be misleading.

REVIEWERS' COMMENTS:

Reviewer #1 (Remarks to the Author):

The authors have sufficiently addressed my concerns and the manuscript is suitable for publication.

Reviewer #2 (Remarks to the Author):

The authors have made the necessary changes or otherwise addressed the concerns with the exception of point 6 from Reviewer #2. It is not clear from the blot in 5e that there is higher phosphorylation in the shRNA(POP2#1) relative to control. The densitometry provided in the rebuttal is not convincing since it appears to only compare the raw values. This data needs to be normalized to the tubulin loading control, particularly since it looks like there is less total protein in the control lanes.

Reviewer #3 (Remarks to the Author):

The authors have adequately addressed all my concerns.

Reviewer response; NCOMMS-16-21466

Reviewer #1:

In this manuscript, Ratsimandresy and colleagues report the generation of a strain of macrophage-specific transgenic mice that expresses human POP2. Studies with this mouse model and human macrophages revealed that POP2 negatively regulates the priming and activation of NLRP3 and AIM2 inflammasomes. As such, POP2 is a novel class of inflammasome inhibitors that may be employed to target pathophysiological conditions involving excessive inflammation.

Previously, the authors employed a CD68 promoter containing macrophage specific IVS-1 enhancer to express POP1 and POP3 in mice. Here the authors used the same strategy to generate a transgenic mouse strain that expresses POP2 only in monocytes, macrophages and DCs. Similar to their in vitro study published previously, the authors showed that the IL-1 secretion downstream of NLRP3 was reduced using monosodium urate crystals as stimulants in the POP2 transgenic mice, concomitant with reduced neutrophil infiltration. Mechanistically, the inhibition of the NLRP3 inflammasome and AIM2 inflammasome was mediated through the direct binding of POP2 with ASC and the reduction of ASC polymerization and subsequent caspase-1 activation.

Surprisingly, the TNF and IL-6 levels were also reduced upon stimulation with either MSU or LPS plus nigericin, suggesting reduction of NF-kB activation and inflammasome priming by POP2, which is distinct from POP1 and POP3 that do not target NF-kB. The reduced I κ B and IKK activation appears to correlate with the suppressive effects of POP2 on NF-kB activation. The above observations in the transgenic mice were confirmed in human THP-1 cells stably expressing POP2.

Because the expression of POP2 was induced by LPS and anti-inflammatory cytokines in a time-dependent manner, which may permit initial protective immune response while facilitating the resolution of inflammation at a later stage, the authors rationalize that POP2 is an important immuno-regulatory molecule to ensure balanced inflammatory responses.

Overall this manuscript elucidates the in vivo function of an important regulator that may be relevant to the control of inflammation during infections and a number of autoinflammatory disorders. The important findings should appeal to a broad readership.

1) One issue that warrants discussion is the effects of POP2 on other members of the inflammasome family. Because POP2 targets ASC, almost all of the inflammasomes may be negatively regulated, including inflammasomes primarily involved in maintaining homeostasis such as the NLRP6 and NLRP12 inflammasomes. How might POP2 impact intestinal homeostasis involving NLRP6 and NLRP12?

The maintenance of intestinal homeostasis is a very complex process. While NLRP6 and NLRP12 have been implicated to contribute to intestinal homeostasis the precise mechanism, and particularly the involvement of ASC and an inflammasome in general, are not fully understood. While NLRP6 expression in goblet cells is essential for proper mucus secretion and control of intestinal pathogen dissemination as well as prevention of intestinal dysbiosis, it is not well established if this is an ASC-dependent inflammasome-mediated activity. We focused in our study on macrophages and currently have no evidence indicating POP2 expression in intestinal epithelial cells (IECs), which are the main cell types involved in NLRP6 inflammasome function. However, if POP2 were to be expressed in IECs we would expect it to function as an NLRP6 inflammasome inhibitor that affects intestinal homeostasis. The absence of NLRP6 or NLRP12 in DSS-induced colitis causes increased disease severity but multiple inflammasomes within different cell-types (e.g., epithelial, hematopoietic cells) accomplish different, and often complementary functions during mucosal immune responses and may vary in different situations or at distinct stages of the disease. We would expect that constitutive expression of POP2 in this context may have adverse

consequences, such as elevated susceptibility to intestinal pathogens, impaired barrier function and bacterial-driven inflammation and potentially increased susceptibility to colitis. NLRP12 controls a checkpoint of non-canonical NF- κ B activation, intestinal inflammation and tumorigenesis. Also in this context, the requirement for an NLRP12 inflammasome has not been demonstrated. A potential role of the NLRP12 inflammasome in response to *Yersinia pestis* has been proposed and in this context, constitutive POP2 expression may increase the susceptibility to infection, comparable to *Nlrp12*^{-/-} mice.

The need for inflammasome regulation in response to pathogens and its potential role in intestinal homeostasis underscores the importance of the highly regulated expression of POP2 as a late response gene, which may not impact early host defense, but attenuate and resolve excessive or prolonged host inflammatory immune responses. Also, as indicated above, we have not yet established that POP2 is expressed in gut epithelial cells. Planned tissue specific POP2 transgenic mice will support addressing such questions in the future. However, to address the question of effects on other ASC-dependent inflammasomes besides AIM2 and NLRP3, we investigated the response of BMDM to the activation of Pyrin inflammasome with the RhoA inactivating toxin from *Clostridium difficile* TcdB, the activation of NLRC4 inflammasome with recombinant flagellin (Fla) and the activation of NLRP1b inflammasomes with the *Bacillus anthracis* lethal toxin (LeTx). POP2 reduces IL-1 β release in response to the activation of all these inflammasomes, underscoring the broad inhibitory range of POP2. These results have been added as a new Figure 2i in the revised manuscript and shown below).

Reviewer #2:

The present study focuses on the unique role of PYRIN domain-only protein POP2 in inhibiting the inflammasome priming and activation in a POP2-transgenic (Tg) mouse model. POP2 ameliorates the inflammatory response in vitro (BMDM) and in vivo (POP2 Tg mice) upon treatment with LPS or mono-sodium-urate crystals, for PAMP and DAMP signals, respectively. Interestingly, this protection against inflammation also occurs systemically by reducing circulatory levels of IL-1 β and IL-6 and TNF- α . THP-1 monocytes and primary human macrophages also show the significant reduction of inflammatory activation in the presence of POP2. ShRNA against POP2 reverses this process and results in the increased levels of IL-1 β and TNF α . POP2 acts via inhibiting NF- κ B activation and also inhibiting ASC-1 polymerization, thereby preventing Caspase-1 activation and inflammatory cytokines response. This paper represents experimental strategy/tools to elucidate the mechanistic role of POP2 in both in vitro and in vivo conditions. Authors have created a model of POP2 transgenic mouse using a CD68 promoter sequence to study the macrophage inflammasomes.

Major comments:

1) Figure 1I, panel shows MPO activity measured in luminescence units for WT and POP2 Tg mice after LPS treatment. The right panel for POP2 mouse does not show dramatic changes in MPO activity. Did authors check if this particular mouse had similar levels of POP2 expression like other POP2 mice showing lower MPO activity?

Our transgenic mice show variable expression of POP2 and it is correct that POP2 expression levels correlate to its potential of inflammasome inhibition and inhibition of neutrophil recruitment measured by MPO activity. Specifically, in Figure 1i (former figure 1I) the mouse shown in the right panel for POP2 (POP2#2815) had a lower POP2 expression than the other POP2 mouse (POP2#2814), which we quantified by qPCR (see figure below, left graph). However, also this POP2 expressing mouse, POP2#2815, shows significantly less MPO activity compared to wildtype mice, which becomes much more obvious upon displaying the quantified picture expressed as total luminescent counts (see figure below, right graph). In addition we show in Figure 6g that transgenic POP2 expression levels correlate to the inhibition of IL-1 β release, which is a prerequisite for neutrophil recruitment, measured by MPO bioluminescence *in vivo*.

2) In Figure 1K, serum IL-6 level does not seem convincing, shown as significantly higher in WT compared to POP-2. It is possible that POP2 can inhibit only secreted serum IL-1 β and TNF- α .

We speculate that the reason for this response is due to the pleiotropic nature of IL-6. In response to systemic LPS-induced inflammation is produced by numerous cell types but in our transgenic mice expression of POP2 is restricted to the monocyte/macrophage/DC lineage. Nevertheless, the reduction of IL-6 in serum is statistically significant. In addition, compared to systemic IL-6, the local cell responses in the peritoneum display a more convincing POP2-mediated reduction of IL-6 in the peritoneal lavage (Figure 1j).

3a) Figure 2e, expression of Caspase-1 p10 seems similar between WT and POP2.

To address this concern, we quantified the protein bands by densitometry, and show the results of a repeat experiment. Upon quantification POP2 expressing cells show a 37% reduction of Caspase-1 p10 compared to wildtype cells (top experiment, below), which agrees with the reduced IL-1 β observed in POP2 expressing cells. We repeated this experiment several times with very comparable results. Here we show a 54.4% reduction in Caspase-1 p10 from a different experiment (bottom experiment, below). We expect that these differences are a consequence of differences in POP2 expression in our transgenic mice.

3b) Figure 2 h and i represent the levels for IL-1 β and IL-18 in response to various other DAMP and PAMP activator molecules. Those differences under CPPD, Poly dA:dT and MSU, do not appear significantly different in WT compared to POP2 probably due to minimal detection. Authors should show these results in a different scale bar graph.

As suggested, we present data in Figure 2h and 2i (now 2h and 2j) on separate scales, which clearly shows that also CPPD is efficiently blocked by POP2. Similarly, poly(dA:dT) and MSU activation is significantly reduced.

4a) In Figure 3, TNF- α and IL-6 increase in WT and ASC^{-/-} mice compared to POP2 upon LPS treatment. Why was a different time scale has been chosen for LPS (0,2, and 16h) and Pam3CSK4 (0,1 and 2h) treatment experiments?

To better compare POP2 effects on LPS or Pam3CSK4-mediated TNF- α stimulation we repeated the experiment using 0, 1h or 2h stimulation for both stimuli.

4b) Also, levels of these cytokines in panel a (2h), b (16h) and c (2h, Pam3CSK4 treatment), seem to be significantly higher for ASC^{-/-} compared to POP2 mice. Have authors looked into this discrepancy?

This observation is consistent with the literature, since IL-6 and TNF- α are produced in an inflammasome independent manner. Hence, ASC has no effect on their secretion (Nature 430, 213-218; 2004). However, the secretion of IL-6 and TNF- α is regulated by the NF- κ B pathway. Here we show that POP2 affects the NF- κ B pathway and therefore IL-6 and TNF- α secretion.

5) Why are expression levels of p-I κ B α different in POP2 in panel e; (iBMDM) and panel g; BMDM at 1h and 2h time points. In fact, it seems that there is an increased expression of p-I κ B α in POP2 iBMDM after one h of LPS induction, and likewise, an increase in expression of p-I κ B α appears at 2h in POP2 BMDM.

We realized that due to some shifted labeling some of the time points did not match up between iBMDM and BMDM. We corrected the labeling and repeated the experiment. We find that phosphorylation of I κ B α (p-I κ B α) usually peaks at 1 hour in all the cell types that we have used (primary/immortal BMDM or THP-1 monocytes). Significantly, POP2 expression in iBMDM and

BMDM results in decreased p-I κ B α . The revised manuscript contains the blots shown here below: the original iBMDM blot (top) and the revised BMDM experiment (bottom).

6) Results in Figure 5e contradict the statement by authors, "Conversely, THP-1shPOP2#1 and THP-1shPOP2#2 cells showed elevated IL-6 levels in response to MSU crystals and nigericin (Fig. 5d) and displayed increased phosphorylation of I κ B α in response to LPS (Fig. 5e)". In fact, the p-I κ B α expression is same in shRNA ctrl and shRNAPOP2#1 post LPS treatment ruling out the effect of shRNAPOP2 on the phosphorylation of I κ B α .

Since overexpression of POP2 blocks IL-6 release (Figure 5a) we expect that removal of POP2 through shRNA expression would have the opposite effect and therefore increase IL-6 release, which is shown in Figure 5d. This is consistent with an increase in the phosphorylation of the inhibitor of NF- κ B (I κ B α), which results in I κ B α degradation and subsequently the release and nuclear translocation of the transcription factor NF- κ B that is responsible for IL-6 transcription. In order to address the reviewers concern about differences in I κ B α phosphorylation between shRNA^{ctrl} and shRNA^{POP2#1}, we quantified the bands and find increased I κ B α phosphorylation in shRNA^{POP2#1} cells at all time points with the most pronounced difference after 1h of LPS treatment, which revealed a 41.4% increase. The quantification is presented below.

7) Authors have shown the regulatory role of POP2 in modulating inflammation in response to PAMPs activators. However, authors should discuss if macrophages expressing POP2 will be efficient in other infection models.

We focused our manuscript on sterile inflammatory responses, but as suggested, we added these discussion points to the revised manuscript. We compellingly show that POP2 expression inhibits sterile inflammatory responses of macrophages through blocking NF-κB and inflammasome signaling. Consequently, monocyte/macrophage-specific expression of POP2 in transgenic mice ameliorates sterile inflammatory disease. Some of the stimuli we tested include bacterial PAMPs or toxins and we therefore expect that POP2 would also ameliorate inflammatory responses triggered by bacterial infection. In fact, we show that POP2 ameliorates inflammation and lethality in models that recapitulate bacterial infection. While these signaling pathways regulated by POP2 are usually beneficial for host defense, prolonged activation or defects in timely resolution can result in sepsis-like syndromes, which are deleterious for the host. Although we have not proven this, we expect that POP2 expression would provide protection by blocking NF-κB-dependent inflammatory gene transcription and by limiting Caspase-1 activation. However, it could impair early host defense responses, especially Caspase-1-driven pyroptosis and potentially result in elevated bacterial burden. However, this risk is likely to be limited due to low basal expression of POP2. The inducible expression of POP2 as a late response gene may then preferentially dampen the excessive host responses, while enabling early host defense.

Minor comments:

1) In Materials and Methods, P24, it is stated that peripheral blood-derived monocytes were differentiated *in vitro* as with BMDM, from 150ul blood for six days. Is this correct? It is not clear how many cells are used in this experiment?

This is correct. We chose this approach to directly determine protein expression in macrophages without requiring the euthanasia of mice. The amount of blood obtained by mandibular or retro-orbital bleeding is variable. We attempted to the best to normalize blood volume, but the final normalization was done after differentiation and lysis of the total cells followed by immunoblot using Tubulin (see Supplemental Figure 3 and below our response to comment 2b from reviewer 3)

Reviewer #3:

The authors used a POP2 knock-in mouse model to elucidate the *in vivo* protective role of POP2 in damping MAMP and DAMP-induced excessive inflammation. These data were further supported with study results using POP2 stably expressing THP-1 human monocytes. The authors convincingly demonstrated that the protective function of POP2 is mediated by inhibiting inflammasome priming and activation. The study is well designed with proper controls, and the

manuscript is well written. However, interpretation of some of the POP2 expression studies (Fig. 5c, Fig 6 f-g) requires clarification.

1) In Fig. 5c, the transcription of GFP and GFP-POP2 mRNA is under the control of pCMV promoter. How does LPS induce both protein expressions? Are these proteins regulated in the post-translational stage by LPS-induced events?

The CMV immediately early (iE) promoter is responsive to NF- κ B following LPS treatment (Eur. J. Biochem. 271, 1094–1105; 2004). In our stable cells, GFP and GFP-POP2 is driven by the CMV_{iE} promoter and therefore we observed inducible expression of both in response to LPS. We used this approach to reflect the LPS-inducible expression of endogenous POP2.

2a) Similarly, in Fig. 6f and g, the promoter of the POP2Tg is CD68/IVS-1 not the endogenous POP2 promoter. Does the POP2 gene expression in the LPS and nigericin treated BMDM POP2 really reflect endogenous POP2 expression in primates under the treatment condition, or it is likely skewed by the differential activation of CD68/IVS-1 promoter?

Although the kinetics of LPS-driven POP2 expression in human macrophages and in mouse transgenic macrophages are not identical, the observed LPS-inducible nature of the hCD68/IVS-1 promoter allows us to reflect the LPS-responsiveness of endogenous POP2, an aspect which we view as an added strength. In addition, we show that the POP2 induction occurs in both cell types as a late response gene, consistent with the general idea that POP2 may have a role in the resolution phase of inflammasome responses. This further underscores that the regulation of POP2 is important for the consequence of the inflammatory response and may explain why we observe a complex regulation. POP2 is inducible expressed in response to PAMPs and cytokines, but also kept in check by NF- κ B and consequently, POP2-mediated blocking of NF- κ B allows itself to further induce its own expression in a feed forward loop. What is further intriguing is that impaired NF- κ B activity is a characteristic feature of LPS-induced tolerance. Future promoter studies are necessary to precisely define the regulatory elements within the POP2 promoter.

2b) In addition, if POP2 proteins are also regulated in translational and post-translational stages, the observed correlation between cytokine (IL-1beta, TNF-alpha) production and POP2 gene expression may be misleading.

We have some evidence that POP2 is also regulated on post-translational level, as blocking the proteasome elevates POP2 expression. Therefore, as we indicated in the figure legend, the quantification of POP2 expression for correlation to secreted IL-1 β and TNF α , was done by determining POP2 protein in individual mice by immunoblot and densitometric quantification and normalization to tubulin levels. Below is the blot used for the particular experiment shown as Figure 6g, and we now also added this information as a Supplemental Figure 3.

Reviewer response; NCOMMS-16-21466A

Reviewer #1:

The authors have sufficiently addressed my concerns and the manuscript is suitable for publication.

Reviewer #2:

The authors have made the necessary changes or otherwise addressed the concerns with exception of point 6 from Reviewer #2. It is not clear from the blot in 5e that there is higher phosphorylation in the shRNA(POP2#1) relative to control. The densitometry provided in the rebuttal is not convincing since it appears to only compare the raw values. This data needs to be normalized to the tubulin loading control, particularly since it looks like there is less total protein in the control lanes.

We repeated the requested experiment in POP2 silenced THP-1 cells, which now compellingly demonstrates that silencing of POP2 elevates activation (phosphorylation) of I κ B α at the peak at 1 hr after TLR4 activation by LPS. We provide this experiment in the revised Fig. 5e and it is also shown below. We hope that this result satisfies the reviewer.

Reviewer #3:

The authors have sufficiently addressed my concerns and the manuscript is suitable for publication.